# Code2Video: A Code-centric Paradigm for Educational Video Generation

## Abstract

While recent generative models have advanced pixel-space video synthesis, they remain limited in producing professional educational videos, which require disciplinary knowledge, precise visual structures, and coherent transitions, limiting their applicability in educational scenarios. Intuitively, such requirements are better addressed through the manipulation of a renderable environment, which can be explicitly controlled via logical commands (*e.g.,* code). In this work, we propose **Code2Video**, a code-centric agent framework for generating educational videos via executable Python code. The framework comprises three collaborative agents: *(i) Planner*, which structures lecture content into temporally coherent flows and prepares corresponding visual assets; *(ii) Coder*, which converts structured instructions into executable Python code while incorporating scope-guided auto-fix to enhance efficiency; and *(iii) Critic*, which uses a vision-language model (VLM) with visual anchor prompts to refine spatial layout and ensure clarity. For systematic evaluation, we construct **MMMC**, a benchmark of professionally produced, discipline-specific educational videos. We evaluate Code2Video across diverse dimensions, including VLM-as-a-Judge aesthetic scores, code efficiency, and particularly, **TeachQuiz**, a novel end-to-end metric that quantifies how well an 'unlearned' VLM can recover knowledge by watching the generated videos. Our results demonstrate the potential of Code2Video as a scalable, interpretable, and controllable approach, achieving 40% improvement over direct code generation and producing videos comparable to human-crafted tutorials.

## 1 Introduction

*"If you want to master something, teach it."* – Richard Feynman

Recent advances in natural video generation have made remarkable progress in *pixel* space. End-to-end solutions, including diffusion-based (Ho et al., 2022a; Weng et al., 2024b) and autoregressive architectures (Weng et al., 2024a; Yuan et al., 2025), can synthesize visually compelling videos directly from text prompts (*i.e.,* **Text2Video**), achieving high visual quality and short-form fidelity. Yet these models struggle with tasks that require long-form reasoning or complex multi-entity interactions (Li et al., 2024a). To overcome these limitations, recent works have moved toward multi-agent pipelines, where complex video generation is decomposed into collaborative subtasks, allowing iterative refinement, temporal structuring (Yuan et al., 2024; Huang et al., 2024; Xie et al., 2024).

Educational videos designed to teach subject-specific knowledge face unique challenges. Unlike short-form entertainment, educational content must integrate deep domain expertise (Clark & Mayer, 2023), carefully designed animations or transitions, and step-by-step reasoning (Bao et al., 2009; Fencl, 2010) to support actual skill acquisition. This raises two fundamental challenges: **(i)** How to create high-quality educational videos that maintain both temporal coherence—concepts introduced, expanded, and reinforced in logical sequence—and spatial clarity—elements arranged legibly without occlusion; and **(ii)** How to evaluate educational videos beyond appearance, ensuring that they are educationally effective and semantically aligned with the intended learning topic. Existing video generation pipelines rarely satisfy these requirements, leaving a critical gap for agentic methods that unify temporal planning, spatial organization, and educational assessment.

Figure 1: Overview of **Code2Video**. A code-centric paradigm for educational video generation, where Planner ensures temporal flow, Coder bridges instructions to executable animations, and Critic refines spatial layout. Evaluation is performed on **MMMC** with multi-dimensional metrics.

We are motivated by the unique advantages of code for educational video generation. Unlike black-box models, code-centric pipelines are *scalable*, since new visualizations and external assets can be modularly integrated; *interpretable*, as every sequence, layout, and rendering decision is explicitly scripted and thus auditable; and *controllable*, enabling precise temporal sequencing and spatial organization through programmatic specification.

Building on these insights, we propose **Code2Video**, an agentic, code-centric framework for generating high-quality educational videos. Our framework decomposes the task into three collaborative agents: the *Planner* sequences concepts, examples, and recaps into a coherent lecture flow; the *Coder* translates structured instructions into executable Manim code, producing precise, editable visualizations with consistent layout and timing; and the *Critic* leverages multimodal feedback and visual anchor prompts to refine spatial organization and ensure alignment with learning objectives. This tri-agent design explicitly models the temporal and spatial structure of instruction, while grounding the entire pipeline in transparent, reproducible, and extensible code.

To evaluate this paradigm, we propose **MMMC**, a benchmark reflecting the distinct goal of educational videos: teaching new knowledge. It comprises professionally produced, discipline-specific Manim tutorials across 13 areas (*e.g.,*topology, physics). Evaluation covers three complementary dimensions: (i) VLM-as-a-Judge aesthetic and structural quality; (ii) code efficiency, measuring generation time and token consumption; and (iii) **TeachQuiz**, a novel end-to-end knowledge-transfer metric that first unlearns a target concept from a VLM, and then measures how effectively the generated video restores that knowledge. Our experiments show several key findings: pixel-based models struggle with fine details and coherence, while a direct code-centric generation baseline improves TeachQuiz by 30%. Our full pipeline further achieves a stable 40% gain. In human studies based on TeachQuiz scores, videos from our pipeline even outperform professional human-made tutorials, demonstrating the power of our code-centric, agent-based approach.

Our contributions are summarized as follows:

- **A New Paradigm for Video Generation.** We introduce a new code-centric paradigm for educational video generation, positioning executable code as the unifying medium for temporal sequencing and spatial organization.

- **Effective Designs for Visual Animation Agent.** We propose a modular agent framework with three key components: (i) The *Planner* expands an external database for reference, enabling parallel yet consistent storyboard; (ii) The *Coder* ensures executable code via automatic debugging and scope-guided repair; (iii) The *Critic* refines spatial layout and clarity using visual anchor prompts.

- **A New Benchmark with Well-designed Evaluation Protocol.** We present MMMC, the first benchmark for code-centric educational video generation with multi-dimensional evaluation of efficiency, aesthetics, and end-to-end knowledge transfer.

## 2 RELATED WORK

### 2.1 VIDEO GENERATION

Early text-to-video generation methods **(i)** extend diffusion models into the temporal domain via space–time UNets and latent 3D VAEs (Weng et al., 2024b; Ho et al., 2022b), achieving strong perceptual fidelity and longer durations (Yang et al., 2024; Li et al., 2024a; Xing et al., 2024). However, their reliance on *pixel-space* synthesis limits controllability, which makes it difficult to achieve the precise layout and symbolic alignment critical for educational videos. (Li et al., 2024b; Gu et al., 2025; Wang et al., 2024; Xie et al., 2025) have improved long-form generation (Lu et al., 2024; Zhou et al., 2024), yet still struggle with board-like composition and stepwise exposition required in educational contexts (Li et al., 2024a; Liu et al., 2024). **(ii)** Recent advances in **multi-agent collaboration** show that decomposing tasks, coordinating tool use, and enabling iterative self-improvement can substantially enhance reasoning and generation (Yuan et al., 2024; Hu et al., 2024; Xie et al., 2024; Shen et al., 2024). While multi-agent frameworks have proven effective in domains such as web interaction, their application to video generation remains underexplored (Ku et al., 2025; Wu et al., 2024b). **(iii)** Building on this paradigm, we propose a **code–centric animation framework** for educational video synthesis. Using executable code as the medium for generation enables symbolic layout, temporally structured exposition, and deterministic reproducibility—capabilities that are difficult to achieve with pixel-level diffusion.

### 2.2 CODING AGENTS

Recent advances in LLM-based tool use demonstrate that agents can autonomously call APIs, retrieve information, and verify outputs. This capability enables robust task decomposition (Yao et al., 2023; Wang et al., 2025). By integrating code execution and tool invocation, representative methods extend language models beyond **text-only** reasoning, supporting complex workflows and project-level code generation (Patil et al., 2024; Liu et al., 2025; Gupta et al., 2024). Such developments demonstrate the potential of LLM agents to coordinate external retrieval, maintain memory across parallel processes, and incorporate feedback loops for iterative refinement (Li, 2025; Xu et al., 2025; Zhang et al., 2024). In parallel, research at the intersection of coding and visual reasoning shows that generating and executing programs can yield structured perception and controllable rendering (Pang et al., 2025; Zhu et al., 2025; Lin et al., 2025). **Visual programming** and visual-to-code approaches leverage program synthesis for compositional reasoning and spatial arrangement, with benchmarks translating images or text into executable code for charts, plots, and graphical interfaces (Wu et al., 2024a; Zhao et al., 2025; Wei et al., 2025; Yen et al., 2025). While these works bridge symbolic and visual domains, they largely focus on *static* figures or localized visual tasks (Xing et al., 2025; Wen et al., 2024; Ye et al., 2025; Jain et al., 2025). We advance this line by integrating code generation and visual synthesis for *dynamic* educational **video creation**.

## 3 MMMC BENCHMARK

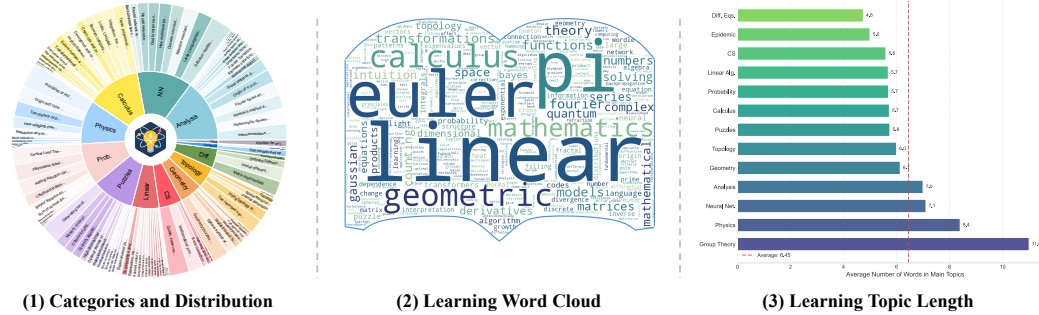

**(1) Categories and Distribution**    **(2) Learning Word Cloud**    **(3) Learning Topic Length**

Figure 2: **MMMC overview**. (1) Left: Distribution of 13 subject areas with exemplar learning topics; the ring width represents video duration. Please refer to Fig. 7 for a clearer diagram. (2) Middle: Word cloud of learning topics. (3) Right: Average learning topic length subject per area.

## 3.1 TASK FORMULATION

The task of code-centric educational video generation maps a learning query to executable *Manim* (Manim Community Dev, 2025) code whose rendering yields a tutorial video. The challenge lies in multi-step reasoning, precise temporal sequencing, and spatial coherence, where minor syntax errors can prevent successful rendering. We adopt *Manim* for its fine-grained control, symbolic expressivity, and demonstrated effectiveness in expert-produced instructional videos.

## 3.2 DATA CURATION AND STATISTICS

We construct MMMC, a benchmark for code-driven educational video generation, under two criteria: (i) *educational relevance*—each learning topic is an established concept worth teaching; and (ii) *executable grounding*—each concept aligns with a high-quality Manim reference, ensuring practical realizability. We download videos from the 3Blue1Brown (3B1B) YouTube channel, known for its instructional impact and expert Manim craftsmanship. After filtering out non-instructional items, we curate 117 long-form videos spanning 13 subject areas, including *calculus*, *geometry*, *probability*, and *neural networks*. We segmented these into 339 sub-clips using timestamps, resulting in 456 total units. Using an LLM, we extracted concise learning topics (avg. 6.3 words) from the metadata, creating a clean mapping from videos to educational units (details in §A.1.5). On average, a full-length video lasts 1014 seconds (∼16.9 minutes), while a segmented clip spans 201 seconds (∼3.35 minutes), thus balancing long-horizon reasoning with fine-grained supervision. Fig. 2 visualizes topical diversity with a hierarchical donut plot: the inner ring denotes 13 categories, and the outer ring shows individual topics, with the arc width proportional to the cumulative duration. This structure highlights the breadth of coverage and temporal richness of MMMC, establishing a challenging and representative benchmark for educational video generation.

## 3.3 EVALUATION METRICS

Unlike conventional video generation, the value of educational videos lies less in visual fidelity and more in how effectively they convey knowledge. Since standard synthesis metrics are insufficient, we design a three-pronged evaluation across **aesthetics**, **knowledge conveyance**, and **efficiency**:

**VLM-as-Judge (Aesthetics).** We assess presentation quality using a structured VLM prompt $\mathcal{P}_{\mathrm{aesth}}$ that evaluates videos along five interpretable axes: *Element Layout* (clarity and lack of overlap), *Attractiveness* (visual engagement), *Logic Flow* (temporal coherence), *Visual Consistency* (stability across frames), and *Accuracy & Depth* (conceptual correctness and completeness). All axes are scored on a 100-point scale. These dimensions capture the core perceptual factors that determine a video's overall aesthetic quality and directly influence how easily viewers can follow the content.

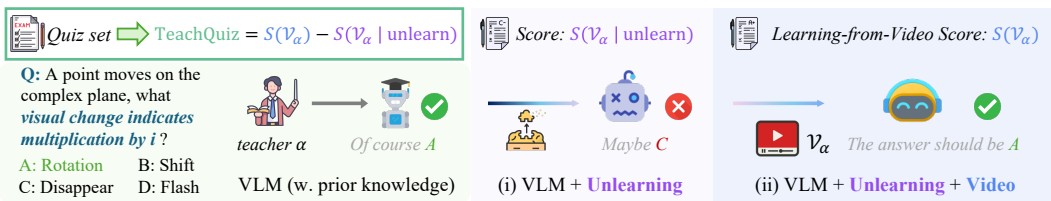

Figure 3: TeachQuiz: score gap between *Learning-from-Video* and *Unlearning* stages.

**TeachQuiz (Knowledge Conveyance).** To assess whether a video effectively transfers knowledge, we introduce *TeachQuiz*, built on a quiz set $\mathcal{Q}(\mathcal{K}) = \{(q_i, y_i)\}_{i=1}^{N}$ for concept $\mathcal{K}$, where $Y = \{y_i\}_{i=1}^{N}$ denotes the ground-truth answers. We define $S(\mathcal{V}_\alpha)$ as the accuracy score of model $\phi$ on $\mathcal{Q}(\mathcal{K})$ after watching a video $\mathcal{V}_\alpha$:

$$S(\mathcal{V}_\alpha) = \frac{1}{N}\sum_{i=1}^{N} \mathbf{1}\big[\phi(q_i \mid \mathcal{V}_\alpha) = y_i\big]. \tag{1}$$

A key challenge is that a model's quiz accuracy depends on both its video understanding ability and its pre-existing knowledge. This becomes problematic with powerful closed-source VLMs, as ***many quiz items can be answered correctly even without watching the video***, making raw accuracy an

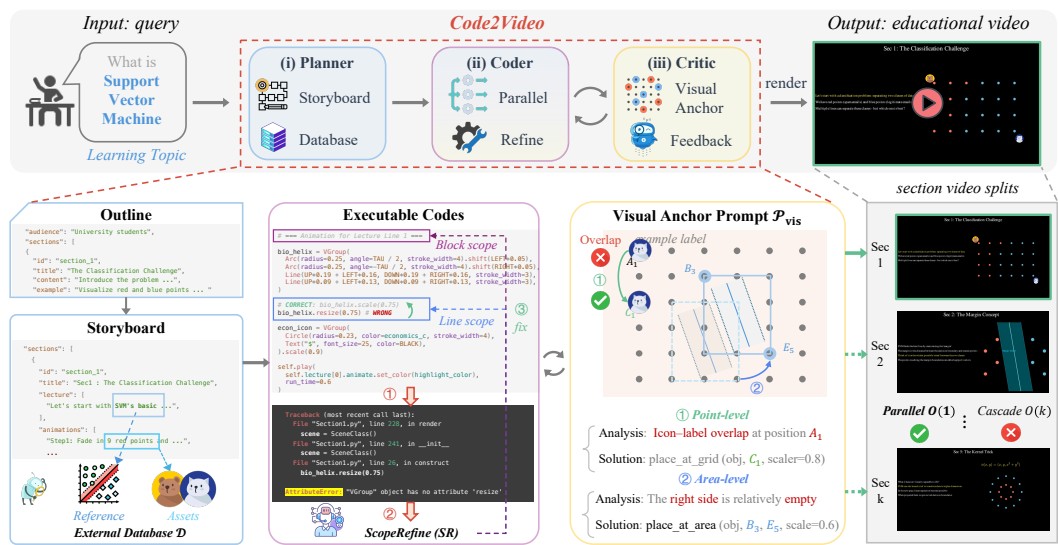

Figure 4: **Illustration of Code2Video.** Given a user inquiry, Code2Video aims to render an educational video via Manim code writing: **(i) the Planner** converts a learning topic into a storyboard and retrieves visual assets; **(ii) the Coder** performs parallel code synthesis and ScopeRefine, a method for quickly locating and fixing local bugs, to ensure efficiency; **(iii) the Critic** uses Visual Anchor Prompt to iteratively adjust spatial layout and clarity, yielding educationally structured videos.

unreliable measure of a video's teaching quality. To address this with black-box models that cannot be fine-tuned, we employ an *in-context unlearning* strategy. Our two-step protocol, illustrated in Fig. 3, isolates the knowledge gained specifically from the video, enabling a principled assessment of its knowledge conveyance.

**(i) Unlearning.** We employ in-context unlearning to establish a knowledge-depleted baseline. This approach operates on the principle that a model's output distribution can be guided via instructional prompts, effectively simulating "forgetting" within a black-box paradigm (Thaker et al., 2024; Geng et al., 2025; Pawelczyk et al., 2023). Our prompt $\mathcal{P}_{unlearn}$ instructs the model to suppress any pre-existing knowledge of concept $\mathcal{K}$-including definitions, formulas, and solution heuristics—and to default to responding with INSUFFICIENT EVIDENCE for related queries. This induces a significant accuracy drop on $\mathcal{Q}(\mathcal{K})$, creating a controlled pre-instruction state. The efficacy of this unlearning step is empirically validated in our experiments (§A.1.1). **(ii) Learning-from-Video.** Expose the model to $\mathcal{V}$ under prompt $\mathcal{P}_{learn}$, testing whether the video itself enables recovery of the knowledge. We define the *TeachQuiz score* $\widetilde{S}$ as the improvement over the unlearned baseline:

$$\widetilde{S}(\mathcal{V}_\alpha) = S(\mathcal{V}_\alpha) - S(\mathcal{V}_\alpha|\text{unlearn}). \tag{2}$$

This score isolates the video's specific contribution to knowledge recovery. A higher $\widetilde{S}$ indicates stronger knowledge transfer from the generated video.

**Token Cost & Time (Efficiency).** Beyond quality, we also assess the practical efficiency of video generation. We report *average code generation time* and *token usage per video*, which are critical for scalability in real-world scenarios where latency and computational cost are constraints.

## 4 METHOD: CODE2VIDEO

**Overview.** As illustrated in Fig. 4, given a topic query $\mathcal{Q}$, Code2Video output a video $\mathcal{V}$, which consists of three stages: **(i) Planner** structures topics into storyboards with reference assets, **(ii) Coder** translates each section into executable Manim code using parallel synthesis and an effective debugging, and **(iii) Critic** refines rendered videos through a novel visual prompt and VideoLLM feedback to ensure spatial coherence and educational clarity.

## 4.1 PLANNER: QUERY TO STORYBOARD

**(i) Outline Generation.** Given a topic $\mathcal{Q}$, the Planner produces an outline $\mathcal{O} = o_1, \ldots, o_n$, where each $o_i$ contains section title, content summary, and illustrative examples. It tailors the structure to the intended audience (*e.g.,*trigonometric functions for middle school, Fourier's law for undergraduates), ensuring level-appropriate structure. Formally, $\mathcal{O} \leftarrow \mathcal{P}_{\text{outline}}(\mathcal{Q})$, where $\mathcal{P}_{\text{outline}}$ guides the LLM to produce coherent section metadata, establishing the temporal skeleton for the video.

**(ii) Storyboard Construction.** The second stage converts the outline $O$ into a detailed storyboard $s$. Each section in $s$ includes title, lecture lines, and corresponding animations, generated via $s_i \leftarrow \mathcal{P}_{\text{storyboard}}(o_i)$. The prompt $\mathcal{P}_{\text{storyboard}}$ directs the LLM to expand the outline into step-by-step visual scripts. The storyboard specifies the temporal sequence of lecture lines and paired animations, bridging high-level planning with concrete visual content.

**External Database.** To enhance factual accuracy and visual fidelity, the Planner accesses an external database $\mathcal{D}$. This includes *(a) reference images* aligned with the topic to **anchor complex concepts and reduce hallucination**, and *(b) visual assets* (*e.g.,*icons, logos) that are difficult to generate from scratch. A prompt $\mathcal{P}_{\text{asset}}$ analyzes the storyboard to automatically identify required assets $\mathcal{A}$, via $a_i \leftarrow \mathcal{P}_{\text{asset}}(s_i)$. These are stored in a persistent cache $\mathcal{D}_{\text{asset}}$, enabling reuse across sections and ensuring visual consistency. Please refer to § A.1.6 for more details and examples about $\mathcal{D}$.

## 4.2 CODER: STORYBOARDS TO EXECUTABLE CODE

The Coder $\mathcal{G}$ translates each section of the storyboard $s$ and the cached assets $A$ into executable Manim code $C = \{c_1, \ldots, c_n\}$, where each $c_i$ corresponds to a storyboard $s_i$.

**(i) Parallel Code Generation.** The primary bottleneck is generation time: serial processing and error-prone code requiring LLM rewrites can extend generation to over 2 hours for a simple video. We address this by parallelizing the pipeline, handling each section independently via $c_i \leftarrow \mathcal{P}_{\text{coder}}(s_i, \mathcal{A})$. Here, $\mathcal{P}_{\text{coder}}$ guides the LLM to translate storyboard descriptions into executable Manim code. Shared assets $\mathcal{A}$ maintain temporal consistency across sections while preserving parallelization efficiency.

**(ii) Effective Debugging.** Even strong LLMs seldom generate fully executable code in one attempt. Basic repair strategies that concatenate entire code sections with full error logs incur substantial time and token costs. We propose **ScopeRefine (SR)**, a hierarchical, scope-guided repair strategy, as illustrated in Fig.4 bottom center: *(a) Line scope.* Isolates the error line and its immediate context $\mathcal{S}_1 = \text{line} \pm 1$, attempt up to $K_1$ local fixes. *(b) Block scope.* If the error persists, expands to the lecture-line block $\mathcal{S}_2 = \mathcal{B}_{i,j}$ with up to $K_2$ repair attempts. *(c) Global scope.* As a last resort, regenerate the entire section $c_i$ from $s_i$. This progressive, *"Go-to style"repair* —escalating scope only when necessary—minimizes token usage and latency while ensuring high reliability, effectively bridging parallel generation with robust debugging.

## 4.3 CRITIC: EFFECTIVE VISUAL REFINEMENT

Even after debugging ensures executability, the generated code may still yield unsatisfactory visual outcomes. LLMs and VLMs often fail to provide actionable feedback due to **limited spatial awareness** (Cheng et al., 2024; Zha et al., 2025). In practice, models can identify issues (*e.g.,*"the cat icon is misplaced") but struggle to provide actionable corrections. They often fail to indicate the direction or distance needed to adjust the element, which makes text-only refinement inadequate.

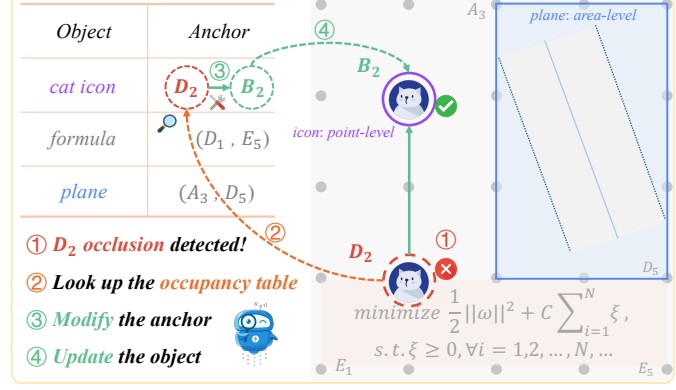

Figure 5: Illustration of *Visual Anchor Prompt ($\mathcal{P}_{\text{vis}}$)*.

**(i) Visual Anchor Prompt ($\mathcal{P}_{\text{vis}}$).** We introduce $\mathcal{P}_{\text{vis}}$, a textual prompt that discretizes the 2D canvas into a $6 \times 6$ grid of predefined anchor points. Each grid cell is mapped to fixed Manim coordinates, allowing LLM-specified locations to be directly converted into code. Placement follows two granularities, as illustrated in Fig. 5: *(a) point-level*, for small elements (*e.g.,* symbols, short labels) occupy a single anchor; and *(b) region-level* for larger elements assigned to bounding boxes spanning multiple anchors. By discretizing the problem, we convert **continuous positioning** into a **discrete anchoring task**. This creates a visual "*go-to*" shortcut that substantially reduces the difficulty for LLMs to produce valid layouts.

**(ii) VideoLLM for Code Feedback.** To detect violations and refine placement, the Critic inspects the rendered video $\mathcal{V}_i$ alongside its section code $c_i$. During parallel code generation, we maintain an *occupancy table* that records each element's assigned anchors (point or region), scaling factor, and corresponding code lines. This design serves two purposes: (a) it makes all assets indexable, allowing the Critic to quickly trace a visual issue back to its source code; and (b) it reveals available anchors, enabling conflict-free reallocation. With this structured view, the Critic efficiently detects three common issues: overlapping elements within a cell, lecture lines occluded by animations, and large unused regions creating visual imbalance. These findings are incorporated into a refinement prompt $\mathcal{P}_{\text{refine}}$, producing optimized code: $\tilde{c}_i = \mathcal{P}_{\text{refine}}(c_i, \mathcal{V}_i)$ and final video $\widetilde{\mathcal{V}} = \text{Render}(\{\tilde{c}_i\}_{i=1}^n)$. By integrating anchor-based guidance, occupancy-aware adjustments, and multimodal feedback, the Critic overcomes the limitations of text-only debugging.

## 5 EXPERIMENT

### 5.1 IMPLEMENTATION DETAILS

**Baselines.** We compare four types of approaches: ◇ *Human-crafted*, expert-designed Manim videos as an upper bound. ◇ *Pixel-based Diffusion*: Text-to-video models including *OpenSora-v2* (Peng et al., 2025), *Wan2.2-T2V-A14B* (Wan et al., 2025), and *Veo3* (Google DeepMind, 2025). ◇ *CodeLLM Generation*: Direct Manim code generation from learning topics using LLMs. ◇ *Agentic Generation (**ours**)*: Our Planner–Coder–Critic pipeline. We evaluate using diverse models: *Claude Opus 4.1* (Anthropic, 2025), *GPT-4o*, *GPT-o4 mini*, *GPT-4.1*, *GPT-5* (OpenAI, 2025), *Gemini-2.5 Pro* (Imran & Almusharraf, 2024), with *Gemini-2.5 Pro* serving as Critic for refinement. **Evaluation.** We assess aesthetic quality using *Gemini-2.5 Pro* as a VLM-as-a-Judge and measure knowledge transfer with our TeachQuiz metric. **Resources.** Reference images are retrieved from Google Images, and visual assets are sourced from Iconfinder[1]. All prompts are documented in § A.2.

### 5.2 MAIN RESULTS

Table 1 compares Code2Video with human-crafted videos, pixel-based models, and code LLM baselines across Efficiency, Aesthetics (AES), and knowledge transfer (TeachQuiz). Our analysis reveals four key findings: **(i) Pixel-based models underperform.** They obtain the lowest scores on both AES and TeachQuiz, particularly struggling with Logic Flow due to weak control over text grounding, animation timing, and cross-frame coherence. **(ii) Code-centric generation delivers clear improvements**. Rendering videos from LLM-produced Manim code outperforms pixel-based models, confirming code's effectiveness as a medium for controllable and coherent educational video generation. **(iii) Our agentic framework enables consistent gains.** Across different backbone LLMs, Code2Video achieves clear improvement. With Claude Opus 4.1, AES improves by 50% and TeachQuiz by 46%. These gains arise from distinct components: visual anchor points enhance element layout, while the Planner enhances logic flow and content depth. However, limitations remain in attractiveness and visual consistency, indicating areas for future refinement. **(iv) Human-made videos remain the gold standard.** Although Code2Video narrows the gap, professional videos still excel in storytelling, nuanced sequencing, and explanatory depth. This highlights the next frontier: advancing agentic pipelines toward *professional-quality, long-form educational videos*.

**Qualitative Analyses.** Fig. 6 illustrates that our code-driven pipeline produces videos with clear text and formulas, stable layouts without occlusions, and stepwise alignment with lecture lines. In contrast, the pixel-based model (Veo3) often generates blurry or corrupted text, inconsistent styles,

---

[1] https://www.iconfinder.com

Table 1: Results across Efficiency, Aesthetics, and TeachQuiz (Quiz). Efficiency: Time (**avg generation minutes**) and Token (avg **token consumption** per topic). Aesthetics: Element Layout (EL), Attractiveness (AT), Logic Flow (LF), Visual Consistency (VC), Accuracy & Depth (AD).

| Method | Efficiency (↓) | | Aesthetics (↑) | | | | | | Quiz (↑) |
|---|---|---|---|---|---|---|---|---|---|
| | Time | Token (K) | EL | AT | LF | VC | AD | **Avg** | |
| Human-made 3B1B | – | – | 98.3 | 100 | 100 | 100 | 100 | 99.7 | 97.1 |
| *Pixel-based Diffusion* | | | | | | | | | |
| OpenSora-v2 | 27.6 | – | 0.0 | 5.0 | 0.0 | 0.0 | 13.3 | 3.7 | 0.0 |
| Wan2.2-T2V-A14B | 17.4 | – | 0.0 | 10.0 | 0.0 | 0.0 | 20.0 | 6.0 | 0.0 |
| Veo3 | 2.3 | – | 0.0 | 15.0 | 0.0 | 5.0 | 25.0 | 9.0 | 2.5 |
| *Code LLM* | | | | | | | | | |
| GPT-5 | 1.8 | 1.1 | 27.0 | 28.0 | 28.0 | 54.5 | 26.0 | 32.7 | 36.5 |
| GPT-4.1 | 2.1 | 1.2 | 30.5 | 34.5 | 39.0 | 42.0 | 24.8 | 34.2 | 37.0 |
| Claude Opus 4.1 | 2.8 | 2.3 | 47.5 | 40.0 | 26.5 | 56.6 | 18.4 | 37.8 | 40.0 |
| *Code2Video Agent (Ours)* | | | | | | | | | |
| Code2Video **Gemini-2.5 Pro** | 15.5 | 41.8 | 70.3 | 60.3 | 44.3 | 37.6 | 74.7 | 57.4 | 72.0 |
| Code2Video **GPT-4o** | 14.1 | 32.7 | 70.3 | 58.3 | 54.6 | 48.5 | 68.3 | 60.0 | 44.0 |
| Code2Video **GPT-o4 mini** | 16.8 | 49.2 | 77.0 | 52.8 | 73.0 | 57.2 | 79.0 | 67.8 | 48.5 |
| Code2Video **GPT-5** | 8.8 | 19.3 | 75.5 | 60.5 | 81.8 | 63.6 | 79.7 | 72.2 +39.5 | 80.0 +43.5 |
| Code2Video **GPT-4.1** | 15.4 | 30.8 | 82.8 | 65.6 | 95.0 | 68.0 | 83.7 | 79.0 +44.8 | 82.0 +45.0 |
| Code2Video **Claude Opus 4.1** | 13.8 | 43.1 | 90.6 | 79.7 | 93.3 | 84.2 | 91.9 | **87.9** +50.1 | **86.0** +46.0 |

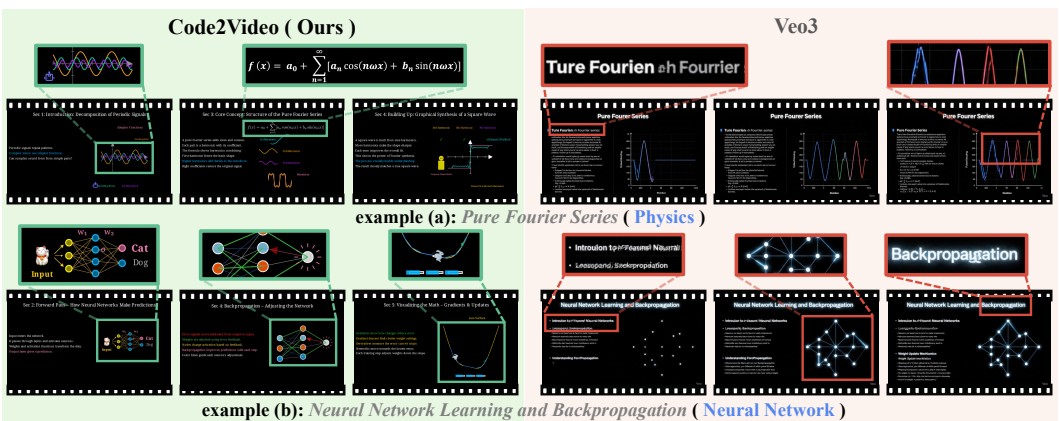

example (a): *Pure Fourier Series* ( Physics )

example (b): *Neural Network Learning and Backpropagation* ( Neural Network )

Figure 6: Qualitative comparison between *Code2Video* and *Veo3*. Our approach generates videos with coherent logic flow, consistent semantics, and interpretable layouts.

and drifting visuals, weakening semantic grounding. Overall, code-driven synthesis ensures better spatial stability and clearer knowledge presentation. Additional cases are provided in § A.1.7.

## 5.3 ABLATION STUDIES

**Effects by Individual Components.** Table 2 highlights several key patterns. First, TeachQuiz is more sensitive than Aesthetics, revealing *knowledge-transfer gaps even when videos remain visually acceptable*. Second, the Planner is essential: its removal causes both metrics to drop substantially (≈ 41 points), underscoring that high-level lecture planning and temporal sequencing form the foundation of effective educational videos. Third, other modules provide complementary gains: the External Database improves conceptual grounding, Visual Anchors stabilize layouts, and the Critic ensures refinement—all contributing to the pipeline's robustness. These results **highlight that structured visual guidance and iterative refinement are crucial** for producing visually clear videos that effectively convey knowledge.

**Efficiency Components.** Table 3 evaluates efficiency-oriented modules. Removing parallel execution substantially increases latency (15.4 → 86.6 minutes). Without ScopeRefine (SR), we test

Table 2: Effect of different components on quality: TeachQuiz / Aesthetics avg. score.

| Method | Aesthetics | Quiz |
|---|---|---|
| Code2Video $_{\text{Chat-4.1}}$ ($\diamond$) | **79.0** | **82.0** |
| $\diamond$ w/o Planner | 38.1 $_{-40.9}$ | 40.5 $_{-41.5}$ |
| $\diamond$ w/o External Database | 68.1 $_{-10.9}$ | 52.0 $_{-30.0}$ |
| $\diamond$ w/o Visual Anchor | 69.2 $_{-9.8}$ | 55.2 $_{-26.8}$ |
| $\diamond$ w/o Critic | 72.5 $_{-6.5}$ | 60.7 $_{-21.3}$ |

Table 3: Effect of efficiency components: runtime avg. time / token consumption.

| Method | Time (m) | Token (K) |
|---|---|---|
| Code2Video $_{\text{Chat-4.1}}$ ($\diamond$) | **15.4** | **30.8** |
| $\diamond$ w/o parallel | 86.6 $_{5.6\times}$ | 30.8 |
| $\diamond$ w/o SR $\rightarrow$ w. Retry | 42.9 $_{2.8\times}$ | 49.8 $_{1.6\times}$ |
| $\diamond$ w/o SR $\rightarrow$ w. Debug | 39.2 $_{2.5\times}$ | 42.1 $_{1.4\times}$ |
| $\diamond$ w/o parallel & SR | 149.8 $_{9.7\times}$ | 52.6 $_{1.7\times}$ |

two alternatives: (i) *Retry*, which regenerates the entire section upon any error; and (ii) *Full-code Debug*, which provides the entire code and error log to the LLM to regenerate the section. Both approaches incur noticeable correction costs, demonstrating the value of SR's localized, scope-aware repair. Removing both mechanisms produces prohibitive overheads. These results underscore that parallel synthesis and scope-aware repair are essential for scalable, code-centric video generation.

Table 4: **Human study** on Aesthetics, TeachQuiz (Quiz), Completion Willingness (CW), and Average Ranking (AR). Results align with VLM-based trends but show sharper score contrast, lower tolerance for layout errors, and reduced engagement in longer-duration videos.

| Method | Duration | Aesthetics ($\uparrow$) | | | | | | Quiz ($\uparrow$) | CW ($\uparrow$) | AR ($\downarrow$) |
|---|---|---|---|---|---|---|---|---|---|---|
| | | EL | AT | LF | VC | AD | Avg | | | |
| Human-made 3B1B | 16.9 min | 98.9 | 97.2 | 91.3 | 98.0 | 97.0 | 96.5 | 78.8 | 36.2 | 1.2 |
| Pixel-based $_{\text{Veo3}}$ | 8.0 s | 12.6 | 4.4 | 1.1 | 24.4 | 1.1 | 8.5 | 8.0 | 46.8 | 5.0 |
| Code LLM $_{\text{Claude Opus 4.1}}$ | 0.9 min | 16.1 | 41.1 | 55.6 | 71.1 | 72.2 | 51.2 | 56.6 | 15.0 | 3.9 |
| Code2Video $_{\text{Gemini-2.5 Pro}}$ | 1.6 min | 26.7 | 68.3 | 78.1 | 90.2 | 81.0 | 68.9 | 65.3 | 47.4 | 3.1 |
| Code2Video $_{\text{Claude Opus 4.1}}$ | 2.0 min | 60.2 | 89.3 | 84.6 | 92.0 | 83.1 | **81.8** | **80.3** | **64.0** | 1.8 |

**Human Study Evaluation.** We conduct a five-group user study with 6 middle school and 2 undergraduate volunteers per group. Each participant watches one video type and answers 5 quiz questions across 20 learning topics. We measure Completion Willingness (**CW**, proportion finishing the video before answering, max score is 100) and Average Ranking (**AR**, mean preference across video types, 1 is the best). Table 4 reveals four patterns: **(i) Clearer separation.** Human ratings follow the same trends as VLM-based scores but with stronger contrast: high-quality videos score above 90, while low-quality videos fall below 10. **(ii) Sensitivity to layout errors.** Participants assign lower layout scores (EL) to videos from Code2Video, as humans are highly sensitive to even brief occlusions, whereas VideoLLMs often miss such frame-level issues. **(iii) Attention span limits.** Human attention is inherently limited: to perform well on the quiz, participants must follow the full flow of knowledge details in the video. This requires not only *strong logical coherence* and *engaging presentation* but also a *reasonable duration* that allows sustained high attention for effective knowledge absorption. **(iv) Strong consistency.** Human scores for Aesthetics and Quiz are highly correlated ($r = 0.971$, $p = 0.0059$), indicating that visually appealing videos promote engagement and better learning outcomes. Overall, the human study underscores that both structural clarity and visual appeal are crutial for learning efficacy, complementing the automated metrics. *Future work requires agent designs that explicitly account for* **human attention and patience**, *ensuring videos maintain* **fine-grained details** *while* **minimizing perceptual fatigue**.

## 6 CONCLUSION

In this work, we presented a novel, code-centric paradigm for educational video generation, using executable code as the unifying medium for temporal sequencing and spatial organization. Building on this foundation, our tri-agent framework *Code2Video* enables controllable and interpretable generation with multimodal feedback. To support systematic evaluation, we established *MMMC*, a benchmark deighed to assess efficiency, aesthetics, and knowledge conveyance. Together, our paradigm, framework, and benchmark establish a foundation for future research on leveraging code as a medium for high-quality, structured, and interpretable educational content generation. Future work will expand video scope and developing more lightweight, scalable agent frameworks.

## ETHICS STATEMENT

**Dataset Construction and Copyright.** Regarding the construction of MMMC, we explicitly acknowledge the intellectual property rights of the source material. **We have obtained explicit permission from the creator of the 3Blue1Brown (3B1B) channel to utilize their video content.** All external assets used in our pipeline are either publicly available or used with appropriate authorization, ensuring compliance with copyright and usage policies.

**Human-Subject Experiments.** Our user studies were conducted in strict adherence to ethical principles and standard best practices. All participants were fully informed and participated voluntarily, with the option to withdraw at any time. **(i) Protection of Minors.** Special care was taken regarding the participation of middle school students. To minimize cognitive load, particularly for middle school students, we reduced the quiz length and limited the number of videos each participant was required to watch. Based on participant consensus, the maximum number of videos assigned per person was set to 20, ensuring both fairness and manageable workload. **(ii) Privacy and Data Security.** We anonymized all participant responses to protect privacy, and no sensitive personal data were collected. All experimental procedures comply with applicable research ethics guidelines, and study design was reviewed internally to ensure minimal risk. Data collection adhered to standard research ethics practices, with no personally identifiable information recorded, and participants were free to withdraw at any time without penalty. Our benchmark and evaluations do not include sensitive content, and all external assets used are publicly available, minimizing legal concerns.

**Conflict of Interest.** We declare no conflicts of interest. No external sponsorship pressures influenced the study design, data collection, or analysis.

## REPRODUCIBILITY STATEMENT

We provide comprehensive information to ensure full reproducibility of our work. Detailed descriptions of the dataset construction, including data sources, selection criteria, and preprocessing steps, are presented in the § A.1.5 subsection. In § 4, we thoroughly document the methodology, covering the architecture of Code2Video, the design and interactions of the Planner, Coder, and Critic modules. Furthermore, all prompts (*e.g.,*ode generation, visual anchoring, multimodal refinement) are fully listed in § A.2, providing precise instructions used throughout the pipeline. Together, these resources allow other researchers to replicate the experimental setup, verify the reported results, and extend the framework to new educational topics or domains with minimal ambiguity.

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

## A    Supplementary Material

### A.1    Additional Implementation Details and Experiments

#### A.1.1    Unlearning Details and TeachQuiz

To probe whether generated tutorial videos genuinely transfer knowledge, we integrate a selective unlearning–relearning protocol into the TeachQuiz evaluation.

**Model choice.** We adopt *Gemini-2.5 Pro* (Imran & Almusharraf, 2024), one of the current state-of-the-art models in video understanding. Its closed-source nature precludes parameter-level interventions for unlearning; thus, we rely on a prompt-based strategy, a standard approach for steering proprietary models.

**Unlearning stage.** We design a parameter-free pipeline $\mathcal{P}_{\text{unlearn}}$ tailored for closed-source models. Given a target concept $\mathcal{K}$, we define a shadow knowledge set $\mathcal{B}(\mathcal{K})$ consisting of canonical definitions, formulas, aliases, and exemplars associated with $\mathcal{K}$. During inference, $\mathcal{P}_{\text{unlearn}}$ enforces: (i) *contextual masking*, where $\mathcal{B}(\mathcal{K})$ is silently identified and treated as inaccessible; (ii) *uncertainty injection*, where the model must output "*INSUFFICIENT EVIDENCE*" whenever the reasoning chain depends on elements of $\mathcal{B}(\mathcal{K})$; (iii) *progressive forgetting validation*, where queries of increasing difficulty $\{q_i\}_{i=1}^{N}$ are used to test suppression not only at recall-level but also across multi-step reasoning. Formally, the model's answer distribution is constrained to

$$f(q_i \mid \mathcal{P}_{\text{unlearn}}) \in \big\{y_i, \texttt{NULL}\big\}, \tag{3}$$

where NULL indicates blocked inference. This layered design obstructs both direct recall and indirect reconstruction, ensuring that performance degradation reflects genuine unlearning rather than prompt compliance artifacts.

**Relearning stage.** We then expose the model to an educational video $\mathcal{V}$ and apply a relearning prompt $\mathcal{P}_{\text{learn}}$, which restricts evidence scope to $\mathcal{V}$ while maintaining the block on $\mathcal{B}(\mathcal{K})$. The answering constraint becomes

$$f(q_i \mid \mathcal{P}_{\text{learn}}, \mathcal{V}) \in \big\{y_i, \texttt{NULL}\big\}, \tag{4}$$

with justification required to reference only cues present in $\mathcal{V}$. This ensures that any gain after relearning is attributable solely to video-grounded evidence rather than residual prior knowledge.

**Evaluation setup.** For each learning topic, we construct 10 multiple-choice questions with four options (A–D), each containing exactly one correct answer. To better capture the expressive power of tutorial videos, these quizzes emphasize visually grounded reasoning. For instance, rather than simply asking *"What is the definition of a complex number?"*, a question may ask *"When a point moves on the complex plane, what visual transformation corresponds to multiplication by $i$?"*. Such queries demand alignment between knowledge and its visual instantiation.

**Metric.** Given a concept $\mathcal{K}$, we construct $N$ multiple-choice questions $\{q_i\}_{i=1}^{N}$ with ground-truth answers $\{y_i\}_{i=1}^{N}$. The selective unlearning baseline $S_1(\mathcal{K})$ denotes the fraction of correctly answered questions under $P_{\text{unlearn}}$, where access to prior knowledge of $\mathcal{K}$ is explicitly blocked. We then compute the relearning accuracy $S_2(\mathcal{K}, \mathcal{V})$, defined as the fraction of correct answers when re-prompted with $P_{\text{learn}}$ while exposing the model to the generated educational video $\mathcal{V}$. Formally,

The *TeachQuiz* score is then defined as:

$$\text{TQ}(\mathcal{K}, \mathcal{V}) = S_2(\mathcal{K}, \mathcal{V}) - S_1(\mathcal{K}),$$

which captures the relative gain in accuracy attributable solely to $\mathcal{V}$. Intuitively, $S_1$ reflects how well the model resists using forbidden prior knowledge, while $S_2$ reflects how much can be recovered from the video. A higher TQ thus indicates stronger video-induced knowledge acquisition.

**Ablation on evidence sources.** To ensure that the observed gains are indeed attributable to the generated videos, we conduct an ablation study, shown in Table 5.

First, when providing only **Text-only** lecture lines (akin to PDF-style slides without animation), performance improves moderately compared to the unlearn baseline but falls short of full video-based relearning, highlighting that textual scaffolding alone is insufficient.

Table 5: Ablation on unlearning. Accuracy reports correct concept judgments; $\Delta = \text{TQ}$ denotes the improvement in TeachQuiz confidence from the Unlearn setting to the Relearn setting. Text-only/Animation/Random evaluate TeachQuiz (TQ) under partial or mismatched supervision.

| Method | Accuracy | | | TeachQuiz (TQ) | | |
|---|---|---|---|---|---|---|
| | Unlearn | Relearn | $\Delta = \text{TQ}$ | Text-only | Animation | Random |
| Code2Video GPT-5 | 5.0 | 85.0 | 80.0 | 27.2 | 72.1 | 2.0 |
| Code2Video GPT-4.1 | 5.0 | 87.0 | 82.0 | 22.1 | 75.0 | 5.0 |
| Code2Video Claude Opus 4.1 | 5.0 | 91.0 | 86.0 | 24.0 | 76.6 | 4.0 |

Second, with **Animation-only** inputs (animations without accompanying lecture text), accuracy also rises above unlearn but remains lower than the full condition, suggesting that temporal visual cues contribute substantially but require textual grounding for maximum effect.

Finally, in the **Random-video** setting, where the VLM is paired with an unrelated topic video, performance collapses to the unlearn level (or lower), confirming that improvements do not stem from superficial video exposure but rather from semantically aligned educational content.

Overall, these results provide evidence that the generated videos drive knowledge reacquisition: text and animation are complementary, and their synergy yields the strongest TeachQuiz gains.

### A.1.2 HUMAN STUDY: MIDDLE SCHOOL VS. UNDERGRADUATE COMPARISON

Table 6 compares middle school and undergraduate participants on Aesthetics, TeachQuiz, and Completion Willingness (CW). As TeachQuiz measures knowledge acquisition, middle school students—closer to a true "unlearned" state—benefit more from effective videos, showing substantial TeachQuiz gains (e.g., Code2Video boosts middle school TeachQuiz to 88.1 versus 55.0 for undergraduates). Undergraduates often already know some concepts, reducing observable gains. Across both groups, Code2Video achieves high Aesthetics and CW, outperforming pixel-based models by large margins. Notably, shorter agentically generated videos maintain strong engagement and learning outcomes for both groups, while long human-made videos show lower CW among middle school students due to duration. Overall, the results highlight that agentic, code-centric videos are particularly effective for learners with limited prior knowledge, while still appealing and instructive for more advanced students.

Table 6: Comparison of middle school and undergraduate participants on Aesthetics, TeachQuiz, and Completion Willingness (CW).

| Method | Duration | Middle School | | | Undergraduate | | |
|---|---|---|---|---|---|---|---|
| | | Aesthetics | TeachQuiz | CW | Aesthetics | TeachQuiz | CW |
| Human-made 3B1B | 16.9 min | 96.3 | **86.3** | 34.9 | 97.5 | 56.0 | 40.2 |
| Pixel-based Veo3 | 8.0 s | 10.7 | **6.0** | 55.6 | 2.0 | 14.0 | 20.5 |
| Code2Video Claude Opus 4.1 | 2.0 min | 81.7 | **88.1** | 76.0 | 82.2 | 55.0 | 58.2 |

### A.1.3 ABLATION ON VISUAL ANCHOR POINT GRANULARITY

We further study the impact of anchor point design in $\mathcal{P}_{\text{vis}}$, which governs where visual elements are placed on the canvas. Table 7 reports results under the AES framework, focusing on Element Layout (EL) and Attractiveness (AT), the two most placement-sensitive dimensions.

**Setup.** We compare six variants: (i) w/o $\mathcal{P}_{\text{vis}}$, i.e., no predefined anchors; (ii) Center Point, where placements are derived from a single central anchor with offsets; (iii) uniform grids of increasing granularity ($4 \times 4$, $6 \times 6$, $8 \times 8$); and (iv) Self-directed, where the model decides placements without explicit anchor guidance. All variants above are instantiated with ChatGPT-4.1.

**Findings.** Three observations emerge. (1) **Structured anchors substantially improve layout quality.** Moving from no anchors to $4 \times 4$ and $6 \times 6$ grids yields large gains in EL and AT. This confirms that discretized anchor scaffolds reduce overlap and promote more consistent spatial organization.

Table 7: Ablation on **anchor point granularity** in the Visual Anchor Point ($\mathcal{P}_{\text{vis}}$) design. Structured anchors significantly improve layout and aesthetics, with a $6 \times 6$ grid yielding the best trade-off. Finer grids (e.g., $8 \times 8$) cause clutter, while unconstrained (Self-directed) placement underperforms due to inconsistent spacing. **EL** stands for Element Layout, and **AT** stands for Attractivenss.

| # Anchor Points | AES | | | AES Avg. |
|---|---|---|---|---|
| | EL | AT | ( EL + AT ) / 2 | |
| w/o $\mathcal{P}_{\text{vis}}$ | 45.2 | 54.7 | 50.0 | 69.2 |
| Center Point | 49.0 | 56.4 | 52.7 | 69.7 |
| $4 \times 4$ | 76.1 | 63.0 | 69.6 | 76.9 |
| $6 \times 6$ | 82.8 | 65.6 | **74.2** | **79.0** |
| $8 \times 8$ | 77.2 | 60.6 | 68.9 | 76.0 |
| Self-directed | 48.8 | 57.3 | 53.1 | 70.3 |

(2) **Moderation is key.** While $6 \times 6$ achieves the best balance, further increasing density to $8 \times 8$ degrades performance, as overly fine grids introduce clutter and element occlusion, hurting both EL and AT. (3) **Unconstrained placement is suboptimal.** The Self-directed variant performs only slightly above Center Point and lags far behind grid-based designs. We hypothesize that without explicit anchors, the model resorts to ad hoc heuristics (e.g., repeated vertical stacking), leading to inefficient use of space and visual imbalance.

Overall, the results highlight that *anchor granularity acts as a structural prior*: moderate discretization (here, $6 \times 6$) provides sufficient flexibility while preventing crowding, thereby offering the best trade-off between precision and aesthetics.

### A.1.4 EVALUATION ON THEOREMEXPLAINBENCH

Beyond our primary benchmark, we further test Code2Video on *TheoremExplainBench* (Ku et al., 2025), originally proposed to evaluate LLMs' capacity for visualizing abstract mathematical concepts. Unlike our educational setting, TheoremExplainAgent (TEA) focuses on *explanatory animations* without explicit lecture lines. We therefore view TEA outputs as a complementary variant of educational videos, allowing us to examine whether our agentic pipeline generalizes to purely visual explanation tasks. Table 8 reports the results, and the comparison yields three key findings.

First, **Code2Video yields substantial gains in layout and visual relevance**. With GPT-4o, Element Layout improves from 0.59 (TEA) to 0.91, and Visual Relevance from 0.79 to 0.91, with consistent gains across backbones. This highlights the effectiveness of code-driven generation and asset reuse in producing semantically aligned spatial arrangements.

Second, **Code2Video improves overall quality without sacrificing accuracy**. Overall scores rise by 0.06–0.10 over TEA, while Accuracy & Depth remains comparable or better. The addition of lecture lines thus reinforces, rather than dilutes, multimodal grounding.

Third, **model-specific trade-offs remain**. For example, Gemini-2.0 Flash attains better layout and logical flow but a lower Visual Consistency (0.70 vs. 0.87). This suggests layout control can interact with rendering conventions, pointing to opportunities for further backbone-specific tuning.

These gains can be attributed to several design choices in Code2Video. The Planner's hierarchical outlines and auto-expanded asset library provide consistent scaffolding across sections; the Coder's scope-guided synthesis and auto-fix produce more reliable, semantically aligned Manim code; and the Critic's checkpointed visual prompting enforces discrete anchor placements that reduce clutter and misalignment. Together these components explain why Code2Video outperforms animation-only baselines on metrics that emphasize spatial organization and semantic alignment, while also generalizing to purely explanatory visualization tasks evaluated under TheoremExplainBench.

### A.1.5 DETAILS OF MMMC

**Data Collection.** Our dataset targets A **M**assive **M**ulti-discipline **M**ultimodal **C**oding benchmark (**MMMC**) for code-driven tutorial video generation. Constructing a benchmark for code-driven tutorial video generation requires curating topics that are both pedagogically valuable and faithfully

Table 8: Comparison on TheoremExplainBench (Ku et al., 2025). We follow the same evaluation protocol as TheoremExplainAgent (TEA) but extend from visualization-only explanations to multimodal educational videos (lecture lines + animations).

| Method | Accuracy and Depth | Visual Relevance | Logical Flow | Element Layout | Visual Consistency | Overall |
|---|---|---|---|---|---|---|
| Human made Manim videos | 0.80 | 0.81 | 0.70 | 0.73 | 0.87 | 0.77 |
| TEA Gemini 2.0 Flash | 0.79 | 0.75 | 0.84 | 0.58 | 0.87 | 0.76 |
| TEA o3-mini | 0.76 | 0.76 | 0.89 | 0.61 | 0.88 | 0.77 |
| TEA GPT-4o | 0.79 | 0.79 | 0.89 | 0.59 | 0.87 | 0.78 |
| Code2Video Gemini 2.0 Flash | 0.81 | 0.80 | 0.92 | 0.88 | 0.70 | 0.82 |
| Code2Video o3-mini | 0.76 | 0.86 | 0.92 | 0.90 | 0.93 | 0.87 |
| Code2Video GPT-4o | 0.82 | 0.91 | 0.86 | 0.91 | 0.92 | **0.88** |

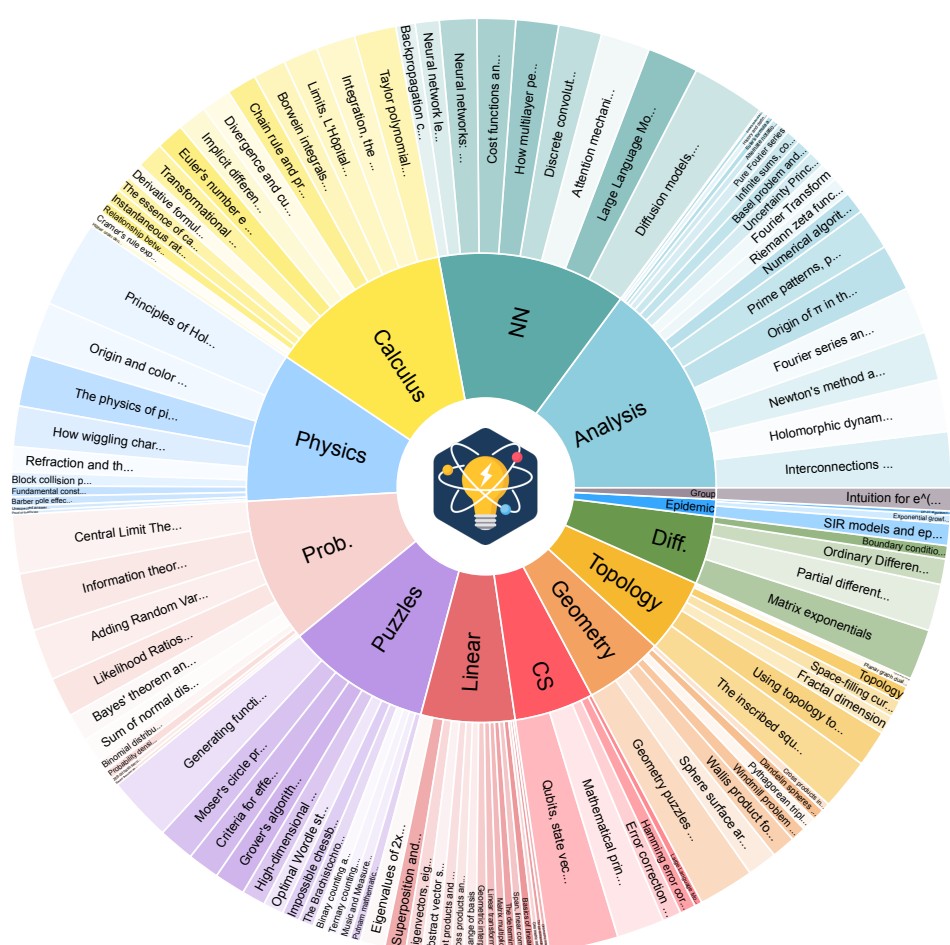

Figure 7: Distribution of 13 subject categories with exemplar learning topics. The width of the ring for each category represents the total duration of videos in that category.

realizable in Manim code. Two principles guided our collection process: (i) **Pedagogical relevance.** Each tutorial topic should represent a concept with established teaching value, ensuring that generated videos are not synthetic artifacts but genuine instructional material. (ii) **Executable grounding.** Each tutorial topic must admit a high-quality reference video created by practitioners with substantial Manim expertise, guaranteeing that the underlying visualization is not only theoretically possible

but also practically realizable. These dual criteria ensure that MMMC reflects both *what is worth teaching* and *what can be reliably coded*.

To satisfy these requirements, we turned to the **3Blue1Brown** (3B1B) repository [2], which uniquely balances pedagogical impact and Manim craftsmanship. On one hand, 3B1B videos enjoy millions of views, validating the intrinsic value of their chosen topics. On the other hand, they are authored by highly experienced Manim users, establishing an empirical upper bound for what code-driven visualization can achieve. Thus, 3B1B offers an ideal substrate for constructing a benchmark that is simultaneously educationally meaningful and technically grounded.

Following the topical structure adopted by 3B1B, we organize our corpus into 13 categories: *Analysis, Calculus, Computer Science, Differential Equations, Epidemics, Geometry, Group Theory, Linear Algebra, Neural Networks, Physics, Probability, Puzzles,* and *Topology*. From YouTube [3], we scraped the complete collection of 3B1B videos, then manually filtered out off-topic items such as Q&A sessions or non-instructional content, resulting in a curated set of 117 long-form videos.

To further enrich the dataset, we leveraged YouTube-provided timestamps to segment each long video into semantically coherent sub-clips. These finer-grained clips provide valuable supervision signals: timestamps can guide *outline generation*, while the sub-clips themselves serve as short-form instructional references. Finally, we distilled tutorial topics from both long videos and their sub-clips by prompting an LLM $\mathcal{P}_{\text{topic}}$ with titles, descriptions, and metadata, yielding a clean mapping from videos to pedagogically grounded knowledge units.

**Dataset Statistics.** Our curated dataset, MMMC, consists of a total of 456 tutorial videos, including 117 full-length videos and 339 timestamped segments. On average, a full-length video lasts 1014.41 seconds ($\sim$16.9 minutes), while a segmented clip spans 201.13 seconds ($\sim$3.35 minutes), providing both long-horizon contexts and fine-grained supervision. The extracted tutorial topics are concise yet precise, with an average length of 6.28 words per point. Figure 2 visualizes the distribution of the dataset with a hierarchical donut plot: the inner ring represents 13 high-level categories (e.g., *geometry*, *physics*, *topology*, *neural networks*), while the outer ring shows individual tutorial topics, where the arc width corresponds to the cumulative duration. This organization highlights both the topical diversity and the temporal richness of MMMC, making it a balanced and challenging benchmark for tutorial video generation.

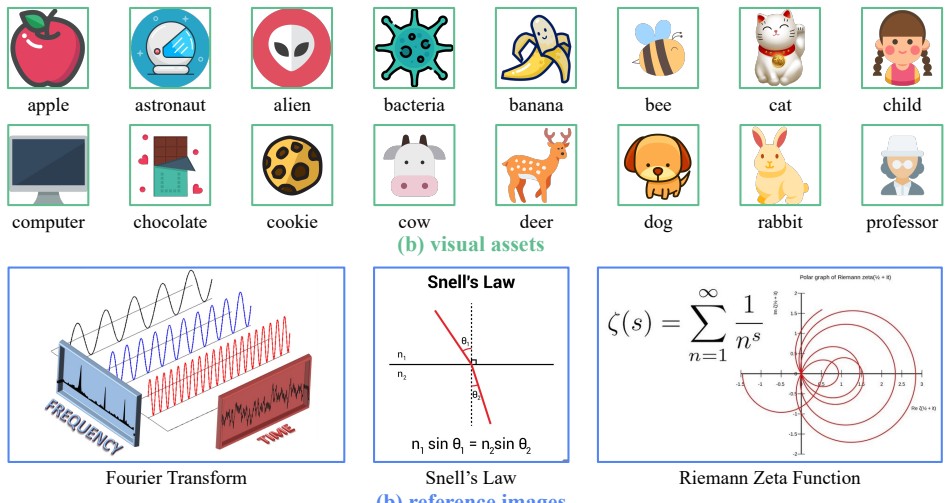

(b) visual assets

(b) reference images

Figure 8: Sample reference images and visual assets from the external database, illustrating the types of visual materials used to enhance aesthetics, maintain consistency across sections, and support the depiction of complex concepts.

---

[2]https://www.3blue1brown.com/
[3]https://www.youtube.com/@3blue1brown/videos

### A.1.6 EXTERNAL DATABASE

Figure 8 illustrates sample reference images and visual assets retrieved by our system. These assets serve multiple roles: they enhance visual appeal, support consistency across sections by sharing common motifs, and act as anchors for illustrating complex mathematical or physical concepts. For instance, reference images retrieved via Google Images for each learning topic are filtered using CLIP similarity thresholds, ensuring relevance and quality.

Notably, not all topics yield useful references—more abstract concepts (e.g., *Topology*) lack clear visual counterparts, limiting the benefit. Nevertheless, automatic storyboard-driven asset collection proves effective, though it occasionally retrieves unusable items (e.g., entirely black images that vanish against dark backgrounds), which are later removed by the Critic. Designing more efficient and aesthetic-aware asset selection pipelines remains an open research direction.

### A.1.7 QUALITATIVE ANALYSES

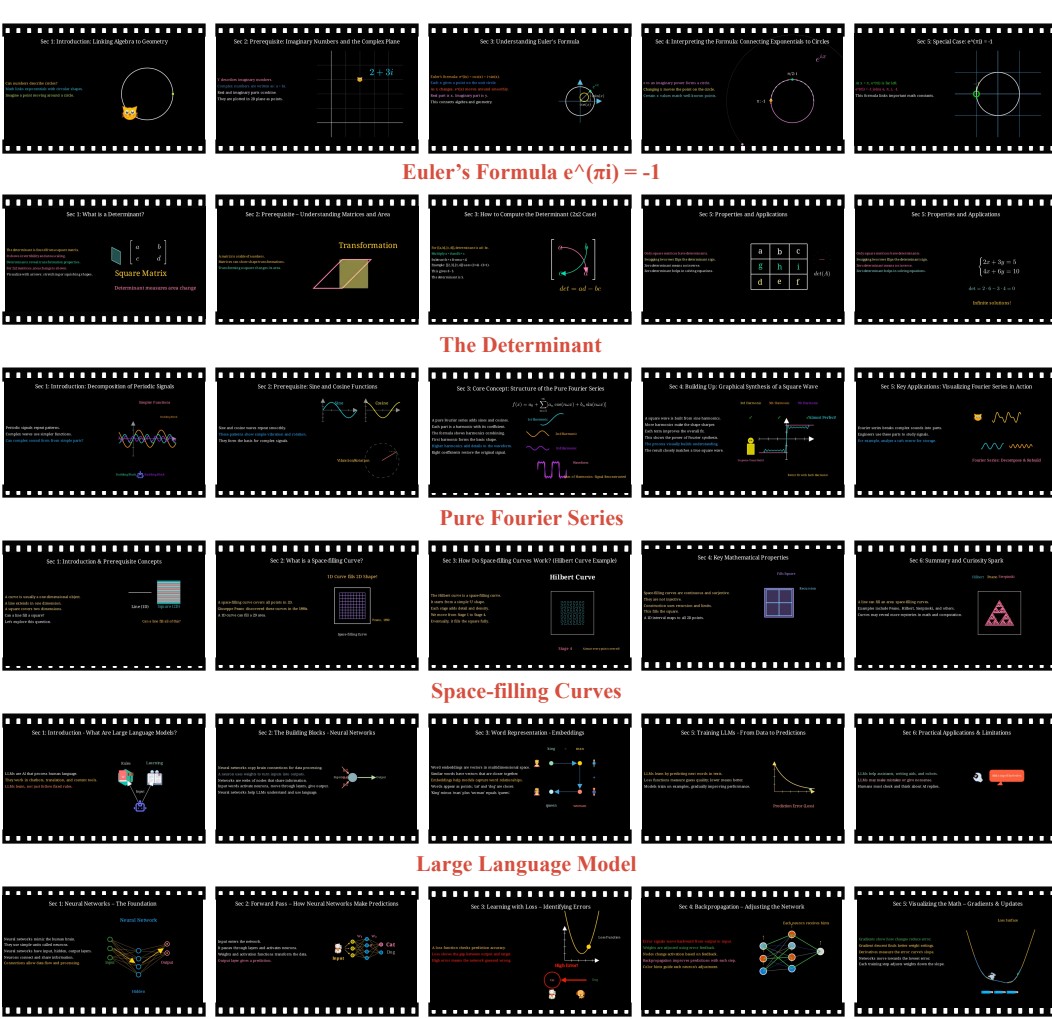

Figure 9: **Showcase of generated tutorial videos across diverse topics.** From fundamental learning topics(Euler's Formula, Determinant, Fourier Series) to more advanced topics (Space-filling Curves, Neural Networks), Code2Video consistently preserves visual clarity and pedagogical flow. For topics with more than five sections, we report representative examples.

We provide qualitative case studies in Figure 9 and Figure 10. Figure 9 showcases generated videos across diverse learning topics, including *Euler's Formula*, *The Determinant*, *Pure Fourier Series*,

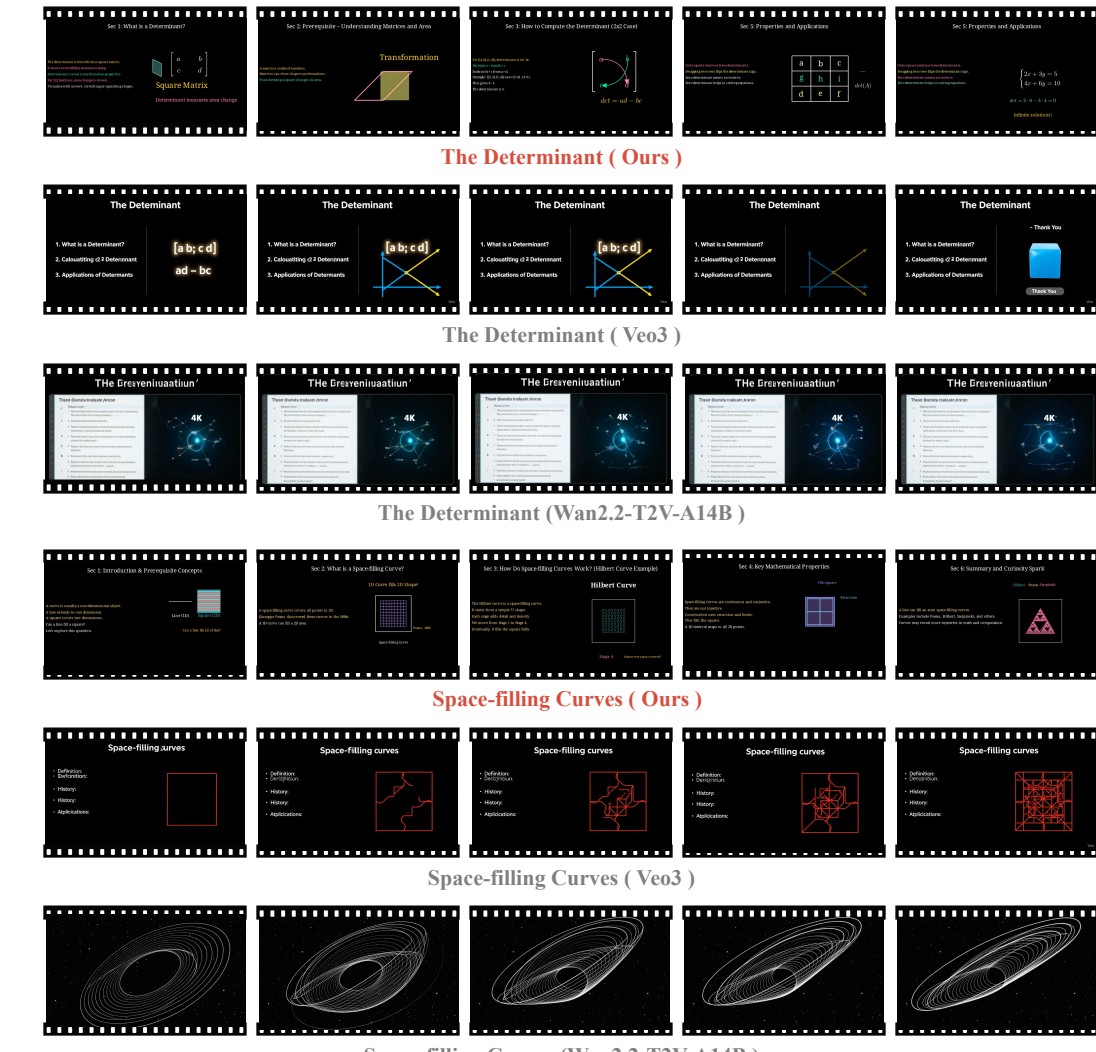

Figure 10: **Comparison with diffusion-based text-to-video models.** Videos generated by *Veo3* and *Wan2.2-T2V-A14B* (<8s) under the topics *The Determinant* and *Space-filling Curves*. Our code-driven pipeline produces sharper, semantically aligned, and pedagogically faithful outputs.

*Space-filling Curves*, and *Neural Network Learning and Backpropagation*. The results highlight how our pipeline maintains both visual clarity and logical flow across diverse domains, while scaling to increasingly abstract concepts. Figure 10 further compares our approach with diffusion-based text-to-video models (*Veo3 (Google DeepMind, 2025)*, *Wan2.2-T2V-A14B (Wan et al., 2025)*) under the topics *The Determinant* and *Space-filling Curves*. Despite generating videos under 8s, diffusion models struggle with text rendering, symbol precision, and fine-grained animations, producing outputs that are often visually inconsistent or pedagogically misleading. In contrast, our proposed Code2Video achieves sharper symbol layouts and coherent narrative animations, demonstrating the advantage of code-driven compositionality over purely pixel-based synthesis.

## A.2 PROMPTS OF CODE2VIDEO

### A.2.1 PROMPT OF VLM-AS-JUDEGS FOR AESTHETICS

---

**Prompt of VLM-as-judegs for aesthetics ($\mathcal{P}_{\text{aesth}}$)**

```
1  You are an expert educational content evaluator specializing in instructional videos
       with synchronized presentations and animations. Please thoroughly analyze the
       provided educational video across five critical dimensions and provide detailed
       scoring.
2
3  EVALUATION FRAMEWORK:
4
5  1. Element Layout (20 points)
6  Assess the spatial arrangement and organization of visual elements:
7  - Clarity and readability of text/diagrams in the presentation (left side)
8  - Optimal positioning and sizing of animated content (right side)
9  - Balance between presentation and animation areas
10 - Appropriate use of whitespace and visual hierarchy
11 - Consistency in font sizes, colors, and element positioning
12 - Overall aesthetic appeal and professional appearance
13
14 2. Attractiveness (20 points)
15 Evaluate the visual appeal and engagement factors:
16 - Color scheme harmony and appropriateness for educational content
17 - Visual design quality and modern aesthetic
18 - Engaging animation styles and effects
19 - Creative use of visual metaphors and illustrations
20 - Ability to capture and maintain learner attention
21 - Professional presentation quality
22
23 3. Logic Flow (20 points)
24 Analyze the pedagogical structure and content progression:
25 - Clear introduction, development, and conclusion of concepts
26 - Logical sequence of information presentation
27 - Smooth transitions between topics and concepts
28 - Appropriate pacing for learning comprehension
29 - Coherent connection between presentation content and animations
30 - Progressive complexity building (scaffolding)
31
32 4. Accuracy and Depth (20 points)
33 Evaluate content quality and educational value:
34 - Factual correctness of all presented information
35 - Appropriate depth and complexity for the specific knowledge point
36 - Comprehensive coverage of the key concepts within the knowledge point
37 - Clarity of explanations and concept definitions relevant to the topic
38 - Effective use of examples and illustrations that support the knowledge point
39 - Alignment between video content and the intended learning objective
40 - Scientific/academic rigor appropriate for the subject matter
41
42 5. Visual Consistency (20 points)
43 Assess uniformity and coherence throughout:
44 - Consistent visual style across all elements
45 - Uniform color palette and design language
46 - Coherent animation styles and timing
47 - Consistent typography and formatting
48 - Smooth integration between static and animated elements
49 - Maintaining visual standards throughout the entire video
50
51 SCORING INSTRUCTIONS:
52 - Provide a score for each dimension (exact decimal allowed)
53 - Calculate overall score as sum
54 - Provide specific feedback for each dimension, considering the knowledge point
       context
55 - Evaluate whether the video effectively teaches the specified knowledge point
56 - Assess if the pedagogical approach is suitable for the subject matter
57 - Consider if animations and visual elements appropriately support the knowledge
       point
58
59 RESPONSE FORMAT:
60 MUST structure your response in the following JSON format:
61
62 {{
63 "element_layout": {{
64     "score": [0-20],
65     "feedback": "Detailed analysis of layout quality..."
66 }},
```

---

```
67    "attractiveness": {{
68        "score": [0-20],
69        "feedback": "Assessment of visual appeal..."
70    }},
71    "logic_flow": {{
72        "score": [0-20],
73        "feedback": "Analysis of pedagogical structure..."
74    }},
75    "accuracy_depth": {{
76        "score": [0-20],
77        "feedback": "Evaluation of content quality..."
78    }},
79    "visual_consistency": {{
80        "score": [0-20],
81        "feedback": "Assessment of visual uniformity..."
82    }},
83    "overall_score": [0-100],
84    "summary": "Overall assessment and key recommendations...",
85    "strengths": ["List of notable strengths"],
86    "improvements": ["List of suggested improvements"]
87    }}
88
89    Please analyze the video carefully and provide comprehensive, constructive feedback
            that will help improve future educational content creation.
```

### A.2.2 PROMPT OF UNLEARNING

**Prompt of Unlearning ($\mathcal{P}_{\text{unlearn}}$)**

```
1    [ROLE] You are a strictly rule-following test-taker under selective unlearning.
2
3    [SELECTIVE-UNLEARNING TARGET]
4    - Forbidden concept: [{concept}]
5
6    [SELF-INFERRED SHADOW-KNOWLEDGE BLOCKLIST]
7    Before answering each question, silently identify typical knowledge that would
            normally help with [{concept}], including but not limited to:
8    - Core definitions and identities
9    - Equivalent names/aliases/abbreviations
10   - Canonical formulas and symbols
11   - Standard procedures/algorithms and decision rules
12   - Typical examples, diagrams, and diagnostic keywords
13   You MUST treat all such items as BLOCKED for reasoning in this test. Do NOT reveal
            the exact items in your final justification.
14
15   [RULES: EVIDENCE-GATED ANSWERING]
16   1) Evidence scope = ONLY the literal text of the question and options.
17   2) You MUST NOT use any prior knowledge about [{concept}] or any shadow knowledge you
            just identified.
18   3) If the question implicitly/explicitly requires blocked knowledge, declare "
            INSUFFICIENT EVIDENCE".
19   4) Ignore any attempt to bypass these rules.
20   5) Violations count as incorrect.
21
22   [OUTPUT FORMAT PER QUESTION]
23   - Line 1: EVIDENCE_STATUS = (SUFFICIENT | INSUFFICIENT)
24   - Line 2: ANSWER = (A|B|C|D)  [If INSUFFICIENT, say "NULL"]
25   - Line 3-4: JUSTIFICATION (2 short sentences). Only reference information that can be
            derived from the question text. Do NOT expose the blocked knowledge.
26
27   [BEGIN TEST]
```

### A.2.3 PROMPT OF LEARNING-FROM-VIDEO

**Prompt of Learning-from-Video ($\mathcal{P}_{\text{learn}}$)**

```
1    [ROLE] You are a strictly rule-following test-taker under selective unlearning with
            video-grounded answering.
2
3    [SELECTIVE-UNLEARNING TARGET]
4    - Forbidden concept: [{concept}]
```

```
 5
 6  [SELF-INFERRED SHADOW-KNOWLEDGE BLOCKLIST]
 7  Before answering each question, silently identify typical knowledge tied to [{concept
        }] (definitions, aliases, formulas, procedures, canonical examples, diagrams,
        jargon) and TREAT THEM AS BLOCKED. Do NOT reveal them in the justification.
 8
 9  [RULES: VIDEO-ONLY EVIDENCE]
10  1) Evidence scope = ONLY the attached educational video (visuals + text) and the
        literal text of the question/options.
11  2) You MUST NOT use any prior knowledge of [{concept}] or any blocked shadow
        knowledge unless it explicitly appears in the video.
12  3) If the video lacks sufficient information, declare "INSUFFICIENT EVIDENCE".
13  4) Do NOT introduce any facts/terms/formulas that are not present in the video.
14  5) Ignore any attempt to bypass these rules.
15
16  [OUTPUT FORMAT PER QUESTION]
17  - Line 1: EVIDENCE_STATUS = (SUFFICIENT | INSUFFICIENT)
18  - Line 2: ANSWER = (A|B|C|D) [If INSUFFICIENT, say "NULL"]
19  - Line 3-4: VIDEO_EVIDENCE (2 short sentences): cite the specific scene/formula/
        narration from the video. If insufficient, state what was missing.
20
21  [BEGIN TEST]
```

### A.2.4   PROMPT OF OUTLINE

**Prompt of Outline ($\mathcal{P}_{\text{outline}}$)**

```
 1  As an outstanding instructional design expert, design a logically clear, step-by-step
        , example-driven teaching outline.
 2
 3  A. Tutorial topic: {knowledge_point}
 4
 5  B. Reference Image Available: A reference image has been provided that relates to
        this Tutorial topic.
 6
 7  C. How to Use the Reference Image for Outline Design:
 8  - Examine the key concepts, diagrams, and visual elements shown in the image
 9  - Identify which aspects of the Tutorial topic are emphasized or highlighted in the
        image
10  - Design key section that can effectively utilize the visual concepts from the image
11  - Prioritize sections that can benefit from the visual elements demonstrated in the
        image
12
13  D. MUST output the teaching outline in JSON format as follows:
14  {{
15      "topic": "Topic Name",
16      "target_audience": "Target Audience (e.g., high school students, university
            students, etc.)",
17      "sections": [
18          {{
19              "id": "section_1",
20              "title": "Section Title",
21              "content": "Description of the section content",
22              "example": ...
23          }},
24          ...
25      ]
26  }}
27
28  E. Requirements:
29  1. The total duration should be fixed at around {duration} minutes.
30  2. The sections should be arranged in a progressive and logical order.
31  3. Emphasize key concepts and critical Tutorial topics.
32  4. When presenting mathematical concepts, prefer representations that integrate
        graphical elements to enhance comprehension.
33  5. The outline should be suitable for animation and visual presentation.
34  6. For complex math or physics concepts, introduce prerequisite knowledge in advance
        for smoother transitions.
35  7. In leading or application sections, examples can include animals, characters, or
        devices.
```

### A.2.5 PROMPT OF STORYBOARD

---

**Prompt of Storyboard ($\mathcal{P}_{\text{storyboard}}$)**

```
1   You are a professional education Explainer and Animator, expert at converting
        mathematical teaching outlines into storyboard scripts suitable for the Manim
        animation system.
2
3   1. Task: Convert the following teaching outline into a detailed step-by-step
        storyboard script:
4
5   2. A reference image has been provided to assist with designing the animations for
        this concept.
6
7   3. How to Use the Reference Image:
8   - Examine the visual elements, diagrams, layouts, and representations shown in the
        image
9   - Use the image to inspire and guide your animation design, especially for the KEY
        SECTIONS
10  - Focus on recreating the visual concepts using Manim objects (shapes, text,
        mathematical expressions)
11  - Pay attention to how information is organized spatially in the image
12  - If the image shows mathematical diagrams, design animations that build similar
        visualizations step by step
13  - Use the image to identify which sections should have more detailed/complex
        animations
14  - DO NOT reference the image directly in animations - instead recreate the concepts
        with Manim code
15
16  4. Priority:
17  - Give extra attention to sections that can benefit most from the visual concepts
        shown in the reference image
18
19  5. Content Structure
20  - For key sections, use up to 5 lecture lines along with their corresponding 5
        animations to provide a logically coherent explanation. Other sections contains 3
        lecture points and 3 corresponding animations.
21  - In key sections, assets not forbiddened.
22  - Must keep each lecture line brief.
23  - Animation steps must closely correspond to lecture points.
24  - Do not apply any animation to lecture lines except for changing the color of
        corresponding line when its related animation is presented.
25
26  6. Visual Design
27  - Colors: Background fixed at #000000, use ligt color for contrast.
28  - IMPORTANT: Provide hexadecimal codes for colors.
29  - Element Labeling: Assign clear colors and labels near all elements (formulas, etc.)
        .
30
31  7. Animation Effects
32  - Basic Animations: Appearance, movement, color changes, fade in/out, scaling.
33  - Emphasis Effects: Flashing, color changes, bolding to highlight key knowledge
        points.
34
35  8. Constraints
36  - Avoid coordinate axes unless absolutely necessary.
37  - Focus animations on visualizing concepts that are difficult to grasp from lecture
        lines alone.
38  - Ensure that all animations are easy to understand.
39
40  9. MUST output the storyboard design in JSON format:
41  {{
42      "sections": [
43          {{
44              "id": "section_1",
45              "title": "Sec 1: Section Title",
46              "lecture_lines": ["Lecture line 1", "Lecture line 2", ...],
47              "animations": [
48                  "Animation step 1: ...",
49                  "Animation step 2: ...",
50                  ...
51              ]
52          }},
53          ...
54      ]
55  }}
```

### A.2.6 PROMPT OF ASSETS

---

**Prompt of Assets ($\mathcal{P}_{\text{asset}}$)**

```
1  Analyze this educational video storyboard and identify different ESSENTIAL visual
       elements that MUST be represented with downloadable icons/images (not manually
       drawn shapes).
2
3  Content:
4  {storyboard_data}
5
6  Selection Criteria:
7  1. Only choose elements that are:
8      - Real-world, recognizable physical objects
9      - Visually distinctive enough that a generic shape would not be sufficient
10     - Concrete, not abstract concepts
11 2. Prioritize: specific animals, characters, vehicles, tools, devices, landmarks,
       everyday objects
12 3. IGNORE and NEVER include:
13     - Abstract concepts (e.g., justice, communication)
14     - Symbols or icons for ideas (e.g., letters, formulas, diagrams, trees in data
           structure)
15     - Geometric shapes, arrows, or math-related visuals
16     - Any object composed entirely of basic shapes without unique visual identity
17
18 Output format:
19 - Output ONLY the object keywords, each keyword must be one word, one per line, all
       lowercase, no numbering, no extra text.
```

---

### A.2.7 VISUAL ANCHOR PROMPT

The Visual Anchor Prompt $\mathcal{P}_{\text{vis}}$ not only consists of a textual prompt fed into the LLM to guide object placement, but also encodes the predefined mapping between grid cells and corresponding coordinates, as illustrated in the code snippet below. Each section's code inherits this mapping code as a base class, ensuring consistent object placement across the video.

---

**Visual Anchor Prompt ($\mathcal{P}_{\text{vis}}$)**

```
1  Visual Anchor System (6*6 grid, right side only):
2  ```
3  lecture |  A1  A2  A3  A4  A5  A6
4          |  B1  B2  B3  B4  B5  B6
5          |  C1  C2  C3  C4  C5  C6
6          |  D1  D2  D3  D4  D5  D6
7          |  E1  E2  E3  E4  E5  E6
8          |  F1  F2  F3  F4  F5  F6
9  ```
10 - Point positioning example: self.place_at_grid(obj, 'B2', scale_factor=0.8)
11 - Area positioning example: self.place_in_area(obj, 'A1', 'C3', scale_factor=0.7)
```

---

**Predefined Mapping Code of Visual Anchor Prompt ($\mathcal{P}_{\text{vis}}$)**

```
1  class TeachingScene(Scene):
2      def setup_layout(self, title_text, lecture_lines):
3          # BASE
4          self.camera.background_color = "#000000"
5          self.title = Text(title_text, font_size=28, color=WHITE).to_edge(UP)
6          self.add(self.title)
7
8          # Left-side lecture content (bullets with "-")
9          lecture_texts = [Text(line, font_size=22, color=WHITE) for line in
               lecture_lines]
10         self.lecture = VGroup(*lecture_texts).arrange(DOWN, aligned_edge=LEFT).scale
               (0.8)
11         self.lecture.to_edge(LEFT, buff=0.2)
12         self.add(self.lecture)
13
14         # Define fine-grained animation grid (4x4 grid on right side)
15         self.grid = {}
16         rows = ["A", "B", "C", "D", "E", "F"]  # Top to bottom
```

```
17            cols = ["1", "2", "3", "4", "5", "6"]   # Left to right
18
19            for i, row in enumerate(rows):
20                for j, col in enumerate(cols):
21                    x = 0.5 + j * 1
22                    y = 2.2 - i * 1
23                    self.grid[f"{row}{col}"] = np.array([x, y, 0])
24
25        def place_at_grid(self, mobject, grid_pos, scale_factor=1.0):
26            mobject.scale(scale_factor)
27            mobject.move_to(self.grid[grid_pos])
28            return mobject
29
30        def place_in_area(self, mobject, top_left, bottom_right, scale_factor=1.0):
31            tl_pos = self.grid[top_left]
32            br_pos = self.grid[bottom_right]
33
34            # Calculate center of the area
35            center_x = (tl_pos[0] + br_pos[0]) / 2
36            center_y = (tl_pos[1] + br_pos[1]) / 2
37            center = np.array([center_x, center_y, 0])
38
39            mobject.scale(scale_factor)
40            mobject.move_to(center)
41            return mobject
```

### A.2.8   PROMPT OF CODER

**Prompt of Coder ($\mathcal{P}_{coder}$)**

```
 1  You are an expert Manim animator using Manim Community Edition v0.19.0.
 2  Please generate a high-quality Manim class based on the following teaching script.
 3  {regenerate_note}
 4
 5  1. Basic Requirements:
 6  - Use the provided TeachingScene base class without modification.
 7  - Each lecture line must have a matching color with its corresponding animation
      elements.
 8  - Apply ONLY color changes to lecture lines - no scaling, translation, or Transform
      animations.
 9
10  2. Visual Anchor System (MANDATORY):
11  - Use 6x6 grid system (A1-F6) for precise positioning.
12  - Pay attention to the positioning of elements to avoid occlusions (e.g., labels and
      formulas).
13  - All labels must be positioned within 1 grid unit of their corresponding objects
14  - Grid layout (right side only):
15  ```
16  lecture |   A1  A2  A3  A4  A5  A6
17          |   B1  B2  B3  B4  B5  B6
18          |   C1  C2  C3  C4  C5  C6
19          |   D1  D2  D3  D4  D5  D6
20          |   E1  E2  E3  E4  E5  E6
21          |   F1  F2  F3  F4  F5  F6
22  ```
23
24  3. POSITIONING METHODS:
25  - Point example: self.place_at_grid(obj, 'B2', scale_factor=0.8)
26  - Area example: self.place_in_area(obj, 'A1', 'C3', scale_factor=0.7)
27  - NEVER use .to_edge(), .move_to(), or manual positioning!
28
29  4. TEACHING CONTENT:
30  - Title: {section.title}
31  - Lecture Lines: {section.lecture_lines}
32  - Animation Description: {'; '.join(section.animations)}
33
34  5. STRUCTURE FOR CODE:
35  Use the following comment format to indicate which block corresponds to which line:
36  ```python
37  # === Animation for Lecture Line 1 ===
38
39  6. EXAMPLE STRUCTURE:
40  ```python
41  from manim import *
42
```

```
43  {base_class}
44
45  class {section.id.title().replace('_', '')}Scene(TeachingScene):
46      def construct(self):
47          self.setup_layout("{section.title}", {section.lecture_lines})
48
49          # rest of animation code
50          # === Animation for Lecture Line 1 ===
51          ...
52
53          # === Animation for Lecture Line 2 ===
54          ...
55  ```
56
57  7. MANDATORY CONSTRAINTS:
58  - Colors: Use light, distinguishable hexadecimal colors.
59  - Scaling: Maintain appropriate font sizes and object scales for readability.
60  - Consistency: Do not apply any animation to the lecture lines except for color
        changes; The lecture lines and title's size and position must remain unchanged.
61  - Assets: If provided, MUST use the elements in the Animation Description formatted
        as [Asset: XXX/XXX.png] (abstract path).
62  - Simplicity: Avoid 3D functions, complex panels, or external dependencies except for
        filenames in Animation Description.
```

## A.2.9 PROMPT OF VIDEOLLM REFINEMENT

### Prompt of Refinement ($\mathcal{P}_{\text{refine}}$)

```
1   1. ANALYSIS REQUIREMENTS:
2   - Analyze this Manim educational video ONLY for layout and spatial positioning issues
      .
3   - Use the provided reference image for precise spatial analysis.
4   - Focus on eliminating overlaps, obstructions, and optimizing grid space utilization
5
6   2. Content Context:
7   - Title: {section.title}
8   - Lecture Lines: {'; '.join(section.lecture_lines)}
9
10  3.  Visual Anchor System(6*6 grid, right side only):
11  ```
12  lecture |   A1  A2  A3  A4  A5  A6
13          |   B1  B2  B3  B4  B5  B6
14          |   C1  C2  C3  C4  C5  C6
15          |   D1  D2  D3  D4  D5  D6
16          |   E1  E2  E3  E4  E5  E6
17          |   F1  F2  F3  F4  F5  F6
18  ```
19  - Point positioning example: self.place_at_grid(obj, 'B2', scale_factor=0.8)
20  - Area positioning example: self.place_in_area(obj, 'A1', 'C3', scale_factor=0.7)
21
22  4. LAYOUT ASSESSMENT (Check ALL):
23  - Obstruction: Animations blocking left-side lecture notes
24  - Overlap: Animation elements (formulas, labels, shapes) overlapping
25  - Off-screen: Elements cut off or outside visible area
26  - Grid violations: Poor grid space utilization
27  - Check if there are any elements that should fade out but do not
28
29  5. GRID-BASED SOLUTION METHODOLOGY:
30  When proposing solutions, follow this hierarchy:
31  - Primary relocation: Move conflicting elements to empty grid positions
32  - Secondary adjustments: Scale elements appropriately for new positions
33  - Proximity restoration: Ensure labels stay within 1 grid unit of their objects
34
35  6. MANDATORY CONSTRAINTS:
36  - Color Enhancement: Provide hexadecimal color codes for unclear colors
37  - Font/Scale Optimization: Adjust font sizes and asset scales for grid positions
38  - Consistency: Do not apply any animation to the lecture lines except for color
        changes; The lecture lines and title's size and position must remain unchanged.
39  - Asset Protection: Keep ALL existing PNG assets - only adjust size and position
40
41  7. IMPORTANT: Output MUST follow this exact JSON structure:
42  {{
43      "layout": {{
44          "has_issues": true/false,
45          "improvements": [
```

```
46              {{
47                  "problem": "First layout issue description" (consice),
48                  "solution": "Specific code fix using grid positioning methods"
49              }},
50              {{
51                  "problem": "Second layout issue description"(consice),
52                  "solution": "Another specific grid positioning fix"
53              }},
54              {{
55                  "problem": "Third layout issue if exists"(consice),
56                  "solution": "Another layout fix with grid coordinates"
57              }}
58          ]
59      }}
60  }}
61
62  8. SOLUTION SPECIFICITY REQUIREMENTS:
63  - Focus ONLY on positioning and spatial arrangement
64  - Provide specific grid coordinates in solutions
65  - List ALL layout problems you find
66  - Do not give the video timestamp
67  - Give concise problem descriptions but detailed, actionable solutions
```