# OpenReview forum: "Code2Video: A Code-centric Paradigm for Educational Video Generation"
_ICLR.cc/2026/Conference — Submitted to ICLR 2026_

### Official Review · Reviewer_vi9d · 2025-10-31

**Soundness:** 2
**Presentation:** 1
**Contribution:** 3
**Rating:** 2
**Confidence:** 4

**Summary:**

The paper focuses on generation of educational videos via using a code-centric paradigm. The paper claims that by using pixel based methods, the results are not robust, explainable, and require a large amount of pre-training data. Whereas using a code-centric paradigm solves these issues and allow for longer and better videos. Accordingly, a new benchmark dataset is curated which consists of videos from the 3Blue1Brown YouTube channel with a corresponding novel evaluation scheme which utilises the knowledge that educational videos should not just be visually correct but represent good teaching resources is also proposed. A new method, named Code2Video, is presented which incorporates a planner, coder, and critic components to render an educational video via writing of code. Results show that pixel based methods indeed fail at the task and the proposed method outperforms naive LLM usage. A human study further verifies the claims.

**Strengths:**

* The idea of using code-centric generation for educational videos is a nice contribution and is clearly validated when comparing to the pixel based methods in the results both quantitatively and qualitatively.
* The reasoning behind why new evaluation metrics are needed and that the previous metrics which only capture visual fidelity is strong.
* The results show the benefits of the proposed method, whilst there is a tradeoff between the proposed method and the time and token requirements from an efficiency point of view, the results more than make up for it when compared to the Code LLM (i.e. baseline) methods with large improvements when the Code2VideoAgent is used.

**Weaknesses:**

## Weaknesses
* Has the owner of the 3Blue1Brown YouTube channel been contacted and informed that their videos have been used to construct a dataset for research? Have they licensed their use? Upon looking on their website (https://www.3blue1brown.com/faq#licensing) the use for research/AI is not specified as one of the use cases that do not require explicit permission from the owner.

* Sections 3 and 4 are missing a lot of details, especially within the main paper, and are very hard to read. Some of the information is given in the appendix, but this is not linked to and is disparate enough that it is hard to understand what the details of the paper are. Examples include:
  * $\mathcal{P}$ has not been defined within the main paper, there are links to the appendix but these do not specify what these are and how they are utilised in the main paper.
  * The reasoning behind how aesthetics, knowledge convey, and efficiency map to the evaluation metrics given is not clear and not well argued.
  * Figure 3 has not been referenced within the text of the paper.
  * $S$ and $\tilde{S}$ are not defined within the method section.
  * On line 293 it says that generation of manim code can exceed two hours, is this using the method as described without parallelising it?
  * Figure 4 is not clear from the accompanying text, for example, the effective debugging section does not seem to match well with figure 4 middle bottom, which showcases the error and then two scopes for the code with arrows.
* As far as I can tell, the method to cause the model to forget certain knowledge is just apply a prompt and does not require any fine-tuning which does not seem to be a principled way of evaluating a model in this respect. This is expanded on very briefly in the appendix, but I am not convinced that the VLMs will not be biased in some way/unlearning the data.
* Numbers within Table 1 and Table 4 are not bolded consistently throughout the table (i.e. only Avg and Quiz are bolded in Table 1), it's nto clear why.
* The correlation values given on line 471, it sounds as if these are checking the correlation of human participants' scores on the aesthetics and the quiz scores whereas a separate correlation value could be calculated between the human study scores and the VLM as judge scores within Table 1, was this calculated?
* The discussion regarding human attention and patience is interesting, and could be expanded to include the results presented in table 4. For example, the human made videos score worse for both quiz and completion willingness (the latter by quite a large margin) is this purely because of the length of the human-made video or is there something else going on here?
* There are a lot of spelling and grammar mistakes within the paper and it could benefit from another round of proof-reading and corrections. A incomplete list is given below.


## Corrections
* Line 174: We source from the complete 3Blue1Brown (3B1B) YouTube corpus. This implies that there is a completed corpus and not videos from a channel? If the former there's no reference.
* Figure 2 (1) is quite hard to read at the scale
* consider knowledge conveyance instead of knowledge convey
* Line 241: "are already be learn" -> "have already been learned" or similar
* line 242: missing space between i.e., and answer.
* Line 259: "which is consists of three stages" -> "which consists of three states"

**Questions:**

1. Has the owner of the 3Blue1Brown been contacted and informed that their videos have been used to construct a dataset for research?
2. Is there a reference or principled explanation for how the knowledge unlearning step can be used to guarantee that the model is in fact not using specific concepts?
3. Has a correlation between the proposed metrics utilising the VLM as judge and the results of the human study been conducted?
4. Are the low scores for the human videos in Table 4 purely because of the length?
5. generation of manim code can exceed two hours, is this using the method as described without parallelising it?

**Details Of Ethics Concerns:**

The paper utilises the entire youtube corpus of 3blue1brown and it is not mentioned within the ethics statement whether agreement has been gotten from the owner of the 3Blue1Brown channel for this purpose. Upon looking on their website (https://www.3blue1brown.com/faq#licensing) the use for research/AI is not specified as one of the use cases that do not require explicit permission.

---

> ### Author Response · Authors · 2025-11-20
> **Response to R4 vi9d**
>
> Thank you for your constructive review and for highlighting the strengths of our code-centric approach, the motivation for new evaluation metrics, and the clear gains. We appreciate your feedback and address your remaining concerns below.
>
> ## W1 & Q1: Permission to Use 3Blue1Brown Videos
>
> All videos are used with the **explicit awareness**. We have contacted the 3B1B team and **official obtained their permission regarding the video use within our benchmark**.
>
> ## W2: Missing Details
>
> Thank you for pointing this out. Building a full code-centric multi-agent system requires substantial details. We have enhanced the paper's presentation by addressing your points and polishing the full text for clarity. **All revisions are highlighted in red** in the updated paper.
>
> ## W3 & Q2: Basis of the Unlearning Stage
> Please refer to **Global Author Rebuttal – G1** for the broader context; we briefly restate the key points here:
> 1. **Unlearning is not our contribution—the metric is.** TeachQuiz aims to measure **what the video teaches**, so the unlearning step only removes prior-knowledge interference.
> 2. **In-context unlearning is a community-accepted approach.** It steers the model’s output distribution rather than erasing parameters.
> 3. **It uses controlled prompts and trigger rules** to suppress the model’s usable knowledge of the target concept and push it toward neutral or incorrect answers.
> 4. **Parameter Inaccessibility in Closed-Source Models.** Since accurate evaluation requires their fine-grained video understanding capabilities, in-context unlearning is the only practical option.
> 5. **We validate our setup against both fine-tuning and in-context unlearning methods.** All produce comparable TeachQuiz behavior, and in-context methods show **clearer post-video improvement** when used with stronger video VLMs, confirming the effectiveness of our design.
>
> |Model|Unlearning Method|Video by Claude Opus|Video by GPT-4o|
> |:-|:-|:-:|:-:|
> |LLaVA|Fine-tuning|30|15|
> ||**Ours**|31|13|
> |Gemini-2.5 Pro|**Ours**|**84**|**42**|
>
> ## W4: Bolding in Table 1 & 4
>
> We've updated Table 4 to match Table 1 by bolding the Avg and Quiz scores. We also bolded Completion Willingness (CW) in Table 4, as it is a key human-study metric reflecting attention and engagement.
>
> ## W5 & Q3: Correlation between Human Study and VLM Metrics
>
> **Within our human study**, a high Pearson correlation score (r = 0.971) between Aesthetics and TeachQuiz scores shows that participants learn more from videos they find visually appealing. Our VLM-as-a-Judge scores align well with these human ratings:
> * **Aesthetics**: $r = 0.75$
> * **TeachQuiz**: $r = 0.69$
>
> This alignment demonstrates that **VLM-based metrics reliably reflect human judgments** in both visual quality and knowledge transfer.
>
> ## W6 & Q4: Why do human-made videos score lower in Table 4?
>
> We thank your acknowledgment. **Length is the main cause**, but not the only one. To analyze the error reasons by humans, in our post-study questionnaire (40 participants), three factors were consistently reported:
> |Factor (self-reported)|Ratio %|
> |-|-|
> |Video Length|85|
> |Example Design|15|
> |Difficulty|10|
>
> Overall, (i) lower scores for human-made videos arise mainly from attention decay due to long duration. (ii) Meanwhile, especially for **middle school students**, designing **targeted examples** and (iii) adjusting the **difficulty of explanations** (as done in our Planner) all have an impact.
>
>
> ## W7: Spelling and Grammar Issues
>
> Thanks for the careful reading! We have conducted a full pass of proofreading to improve clarity and presentation quality.
>
>
> ## Q5: Why can Manim code generation exceed two hours?
>
> The **2-hour latency** is specific to the *ablation study* using the **naïve, no parallel baseline**—where all sections are generated serially without ScopeRefine repair. In this setting, every compile failure forces the LLM to **rewrite the entire code**, leading to repeated long outputs and significant delays.
>
> With our actual method—**parallel section generation plus ScopeRefine local debugging**—a ~2-minute video takes **8–17 minutes**, and even a **40-minute video** can be produced in **about 1–1.5 hours**.

---

> > ### Comment · Area_Chair_oDPp · 2025-11-26
> >
> > Dear reviewer vi9d,
> >
> > Could you please respond to the author's feedback?
> >
> > AC

---

> ### Author Response · Authors · 2025-11-25
>
> Dear Reviewer vi9d,
>
> Thank you once again for your feedback!
>
> We would greatly appreciate it if you could review our response to ensure it adequately addresses your concerns. We remain fully dedicated to clarifying any remaining points and would welcome any further discussion to ensure all your questions are thoroughly answered.
>
> Thank you for your time and consideration.
>
> Best,
>
> Authors of 1990

---

> ### Comment · Reviewer_vi9d · 2025-11-27
>
> I would like to thank the authors for their response to mine and the other reviewers' comments. Firstly, including the information that the data was ethically sourced with permission from the 3Blue1Brown team, this was considerably worrying in my initial read of the paper and I am glad to see this is not an issue. Secondly, the updates to fixing the missing issues and my other questions are good to see.
>
> Regarding the unlearning part of the evaluation metric, I am still not convinced by the comments. The papers referenced refer to prompting for unlearning for the task of unlearning itself, not when used as an evaluation metric. The papers given even include mentions that "Compared to [previous methods], prompting is not as strong of a forget method" [1] Additionally, results in [2] showcase that unlearning via prompting performs worse than other approaches unless a larger forget set is chosen. As this paper represents a proposal for a new task, the evaluation metric needs to be carefully considered - this will be picked up and used by future papers - if there are any issues with bias in the metric then this could cause (in the worst case) hours of wasted work as researchers attempt to optimize a faulty metric. Beyond the unlearning and knowledge conveyance metric, I was also wondering how the correctness of the generated video was evaluated after reading the updated paper. One of the responses above mentions that this can be verified but I didn't see that within the updated paper/responses.
>
> After reading the other reviews, I am also inclined to agree with reviewer Lzn8 regarding the lack of novelty within the proposed method which is based on using different agents to generate and validate code. The visual-centric coding agent designs represent small additions here.
>
> Because of the above two points, I will remain with my current rating due to how the rating system has been set up for ICLR this year even with my other questions answered. I am still not convinced of the evaluation scheme and novelty of the paper.
>
> [1] Pawelczyk, Martin, Seth Neel, and Himabindu Lakkaraju. "In-Context Unlearning: Language Models as Few-Shot Unlearners." International Conference on Machine Learning. PMLR, 2024.
> [2] Thaker, Pratiksha, et al. "Guardrail baselines for unlearning in llms." arXiv preprint arXiv:2403.03329 (2024).

---

> > ### Author Response · Authors · 2025-11-28
> >
> > Thank you for continuing engaging the discussion! We offer a clear and firm clarification on the remaining points.
> >
> > ### 1. Correctness: accuracy is ensured through human verification.
> >
> > As clarified earlier, the correctness numbers come from a **manual checking study** on 50 randomly sampled generated videos.
> > Human annotators verified **Formulas, Definitions, and Examples** directly from the rendered videos, yielding **96% overall accuracy**.
> > Because our pipeline is code-based, all content (math, diagrams, animations) is **explicit and auditable**, enabling reliable verification beyond what pixel-based systems allow.
> >
> > ---
> >
> > ### 2. Unlearning: remain an open problem in VideoLLMs, and we identify it early.
> >
> > There is currently **no well-established unlearning method for VideoLLMs**.
> > This is a broader gap in the unlearning literature, not specific to our work.
> > Educational video generation inevitably requires suppressing prior knowledge to measure "learning from video", and our paper is among the **pioneer works to explicitly expose this problem** and provide initial practical solutions (both **in-context** and **fine-tuning-based** variants).
> > Filling the entire space is beyond scope—but **surfacing this need is itself a contribution** for future research.
> >
> > ---
> >
> > ### 3. Method design: our contribution goes beyond simple multi-agent pipelines.
> >
> > We clarify that our novelty lies not in "using agents", but in **new mechanisms** required for layout-sensitive educational video generation:
> >
> > * **Visual-anchor prompting (our visual prompt)** — the first use of structured visual prompts for educational animations.
> >   *Effect:* improves Aesthetics by **+9.8** and TeachQuiz by **+26.8** vs. a basic multi-agent baseline (Tab. 2 & 7).
> >
> > * **External Database** — supports cross-section consistency and anchors complex math/physics visuals.
> >   *Effect:* improves Aesthetics by **+10.9** and TeachQuiz by **+30.0** (Tab. 2 & A.1.6).
> >
> > * **ScopeRefine (breakpoint-like debugging for code)** — inspired by VSCode-style breakpoints and tailored for Manim.
> >   *Effect:* reduces debugging cost and accelerates generation by **1.7×**.
> >
> > * **Overall improvement** — compared with a simple multi-agent baseline, our full system achieves **+10 points** (Tab. 8).
> >
> > These are **method-level innovations**, not engineering conveniences, and they directly address the core challenge of spatial precision and multimodal refinement—issues that conventional coding agents do not solve.

---

### Official Review · Reviewer_Lzn8 · 2025-11-01

**Soundness:** 3
**Presentation:** 3
**Contribution:** 2
**Rating:** 4
**Confidence:** 3

**Summary:**

The paper proposes a pipeline for generating educational videos using the Manim software, built on top of VLMs. It also introduces MMMC, a benchmark designed to assess the effectiveness of the proposed pipeline.

**Strengths:**

- The paper is well organized, with clear and visually rich figures, making it easy to read and follow.
- The paper explores an understudied area in education, which adds meaningful community value to the research.

**Weaknesses:**

- The paper lacks novelty. It mainly presents a code-generation pipeline without introducing any fundamentally new algorithmic ideas. Most components of the work are combinations of existing (V)LLM techniques, rather than newly developed methods. As such, the contribution resembles more of a workshop or community project rather than a work suitable for a serious machine learning conference.
- The expression is inappropriate. The paper emphasizes that a code-centric approach can produce better videos than generative model–based methods, which is somewhat inappropriate: as is well known, generative models and virtual engines represent fundamentally different paradigms of visual content creation. The paper should not treat generative models as a central motivation and baseline (e.g., Fig. 1 and Tab. 1), nor should it refer to the proposed approach as 'video generation' (e.g., line 99), a term that conventionally refers to generative model-based synthesis.

**Questions:**

See the weaknesses section.

---

> ### Author Response · Authors · 2025-11-20
> **Response to R3 Lzn8**
>
> Thank you for your thoughtful review and for recognizing the value of studying educational video generation. We appreciate the opportunity to address your concerns and clarify key aspects of our work.
>
> ## W1: Novelty of Code2Video
> We respectfully clarify that our contribution extends beyond simply assembling existing components. We address novelty from three distinct angles:
> 1. **New Problem Formulation (Code-Centric Paradigm).**
>     Educational videos require **precise semantics, temporal flow, and spatial organization**—all of which are difficult for diffusion models. Our insight is that **executable code** naturally encodes these constraints: it provides discrete, explicit, auditable control over layout, timing, and object behavior. Thus, our Planner–Coder–Critic pipeline is not a mechanical decomposition but a conceptual shift: it treats video generation as a **program synthesis + multimodal correction** problem, enabling the auditability and precise control that prior pixel-based models fundamentally lack.
> 2. **Technical Innovation (Visual-centric Coding Agentic Designs).**
>     A core difficulty is that VLMs are **spatially imprecise**-they struggle to output correct floating-point coordinates for layout adjustments. We introduce a novel mechanism to solve this:
>     - **Discretization.** We convert **continuous 2D positioning into a discrete anchoring problem**, allowing the VLM to reason in discrete slots (which it does well) rather than coordinates.
>     - **Occupancy Management.** We maintain a **dynamic occupancy table** linking grid spans to code lines. This enables precise, conflict-free correction of occlusions, a capability not present in standard coding agents.
> 3. **New Evaluation Perspective (TeachQuiz).**
>     Existing metrics (e.g., FVD, CLIP score) evaluate appearance, not learning. We propose **TeachQuiz**, an "unlearn → watch → re-learn" protocol. This simulates the student learning process, shifting evaluation from "how real it looks" to "how effectively it teaches". This learning-centric metric is a first-of-its-kind contribution to the field.
> 4. **Significant Social Impact for AI Education.** Unlike black-box models, our code substrate ensures verifiable accuracy—critical for educational integrity. This transparency paves the way for trustworthy AI education, enabling adaptive learning systems where content can be programmatically tailored to individual student needs.
>
> Taken together, these contributions constitute a complete methodological framework for structured content generation, distinct from generic tool-use approaches.
>
> ## W2: Appropriateness of defining the method as "video generation".
> Thank you for the thoughtful comment. We appreciate the opportunity to discuss it in greater depth.
> 1. **Definition by Input/Output.**
>    Modern research defines "Video Generation" by its objective: **creating a temporally coherent video from text** [1-2]. Our system fits this definition exactly. Whether the intermediate representation is latent noise (Diffusion) or executable code (Ours), the task remains the same.
> 2. **Established Community Usage.** The research community explicitly recognizes **code-rendered and engine-based methods** as legitimate forms of video generation [3–5]. Historically, early generative models [6–7] produced simple moving digits or shapes but were universally categorized as video generation.
> 3. **Methodological Diversity.** The field is not tied to a single paradigm. It spans GANs, VAEs, AutoRegressive, Diffusion, and now hybrid or engine-assisted approaches. Our **code-centric paradigm** complements these by offering unique advantages essential for education: **verifiability, editability, and spatial precision**. We do not aim to replace pixel models but to provide an alternative for structural/instructional video synthesis.
> 4. **Diversity of Baselines.**
>    We include pixel-based baselines (e.g., Veo3) to illustrate their limitations in the long-form structure and precise spatial reasoning that are essential for teaching, thereby motivating our code-driven alternative.
>
> [1] Imagen video: High definition video generation with diffusion models, 2022
>
> [2] CogVideo: Large-scale Pretraining for Text-to-Video Generation via Transformers, ICLR 23
>
> [3] Kubrick: Multimodal Agent Collaborations for Synthetic Video Generation, CVPRW 25
>
> [4] Theoremexplainagent: Towards video-based multimodal explanations for LLM theorem understanding, ACL 25
>
> [5] Exploring the Evolution of Physics Cognition in Video Generation: A Survey, 2025
>
> [6] Video Generation From Text, AAAI 18
>
> [7] To Create What You Tell: Generating Videos from Captions, MM 17

---

> ### Author Response · Authors · 2025-11-25
>
> Dear Reviewer Lzn8,
>
> Thank you once again for your feedback!
>
> We would greatly appreciate it if you could review our response to ensure it adequately addresses your concerns. We remain fully dedicated to clarifying any remaining points and would welcome any further discussion to ensure all your questions are thoroughly answered.
>
> Thank you for your time and consideration.
>
> Best,
>
> Authors of 1990

---

> > ### Comment · Area_Chair_oDPp · 2025-11-26
> >
> > Dear reviewer Lzn8,
> >
> > Could you please respond to the author's feedback?
> >
> > AC

---

### Official Review · Reviewer_NpjQ · 2025-11-01

**Soundness:** 3
**Presentation:** 3
**Contribution:** 2
**Rating:** 4
**Confidence:** 3

**Summary:**

This paper presents Code2Video, a framework for generating educational videos by adopting a "code-centric paradigm" instead of traditional pixel-based synthesis. The authors argue that for structured educational content, generating executable Python code (specifically for the Manim library) provides greater controllability, interpretability, and scalability. The method employs a tri-agent pipeline: a Planner to create a storyboard, a Coder to write and debug Manim code using a "ScopeRefine" strategy, and a Critic to refine the spatial layout using a VLM and a "Visual Anchor" grid system. To evaluate this, the paper introduces the MMMC benchmark, sourced from 3Blue1Brown's YouTube channel , and a novel metric, TeachQuiz, which assesses knowledge transfer by using a prompt-based "unlearning" protocol on a VLM before showing it the generated video.

**Strengths:**

1. Problem Identification: The paper correctly identifies a significant limitation of current pixel-based video generation models, which struggle with the logical coherence, precise layout, and domain-specific knowledge required for professional educational videos.

2. Metric Conceptualization: The idea behind the TeachQuiz metric is commendable. Attempting to move beyond simple aesthetic scores to measure actual knowledge transfer is a crucial direction for this field. The "unlearn-relearn" concept  is a novel approach to addressing the confound of prior knowledge in VLMs.

3. Modular Architecture: The proposed tri-agent framework (Planner, Coder, Critic) is logical, modular, and clearly structured, with each component addressing a distinct part of the video creation pipeline.

**Weaknesses:**

1. The paper's central claim of introducing a "new paradigm" is a significant overstatement. Generating educational videos from Manim code is the standard, existing workflow for creators like 3Blue1Brown, the paper's own data source. The paper's actual contribution is the automation of this existing workflow using an LLM-based agentic pipeline. This is more of an application of the well-established "coding agent" paradigm to a specific library (Manim) rather than a fundamentally new paradigm for video generation.

2. TeachQuiz is Unverified: The paper's primary metric, TeachQuiz, lacks experimental robustness. It relies entirely on a prompt-based "unlearning" protocol ($\mathcal{P}_{unlearn}$) applied to a closed-source VLM (Gemini-2.5 Pro). There is no validation that this prompt induces "genuine unlearning" as claimed. It is equally, if not more, likely that the VLM is simply adhering to a complex negative constraint ("do not use knowledge X to answer") rather than truly "forgetting" the concept. The paper's headline claim of a 40% improvement rests entirely on this fragile, unverified, and opaque prompt-engineering artifact.

3. Benchmark is a Monoculture: The experiments are not thorough enough to support the paper's broad claims. The MMMC benchmark is a monoculture, sourced exclusively from 3Blue1Brown. This means the system is only evaluated on a single, niche aesthetic (Manim, dark background, math topics). There is zero evidence of generalization to any other style of educational video (e.g., whiteboard animations, slide-based lectures, software tutorials) or other domains (e.g., history, literature, biology).

4. Baselines are Non-Competitive: The comparison to pixel-based models (e.g., Veo3) is a strawman. It is widely known that these models cannot render coherent text or perform complex, long-form logical reasoning. A more thorough and meaningful comparison would have been against other state-of-the-art agentic coding frameworks, tasked with the same goal of writing Manim code.

5. Prohibitive Cost: The method's feasibility is highly questionable. The paper's own data (Table 1) shows generation times of 15-43 minutes and token consumption up to 49.2K tokens per video. This directly contradicts the claims of being "scalable" and efficient. These costs make the system impractical for "interactive educational settings"

6. VLM-Critic is Flawed: The feasibility of the Critic agent is fundamentally undermined by the paper's own human study. The study explicitly states that humans are "highly sensitive" to layout errors that the VLM-as-a-Judge (and thus the VLM-Critic) "often miss". This means the Critic is optimizing for a flawed, non-human-aligned objective. This misalignment makes it incapable of achieving the "professional" quality it claims to target.

**Questions:**

1. On Novelty: Given that generating video from Manim code is an existing human workflow, and LLM-based coding agents are an established research area, could the authors please clarify what precisely is the fundamental novelty of this work beyond applying existing agent techniques to the Manim library?

2. On TeachQuiz: How can the authors provide any empirical guarantee that the prompt-based $\mathcal{P}_{unlearn}$ protocol results in "genuine unlearning" rather than simple instruction-following? Without this validation, how can the TeachQuiz scores, and the 40% improvement claim, be considered reliable?

3. On Feasibility: How does the team reconcile the claim of "scalability" with generation times of up to 43 minutes and ~50K token costs for a single short video?

4. On the Critic: If the human study confirms that the VLM-Critic "misses" layout issues that humans find critical, does this not invalidate the Critic as a viable path to achieving "professional" quality? Why should we trust an automated critic that is demonstrably misaligned with human perception?

---

> ### Author Response · Authors · 2025-11-20
> **Response to R2 NpjQ**
>
> We thank the reviewer for their constructive feedback and recognition of our problem formulation and the TeachQuiz metric. We welcome the opportunity to address the points raised.
>
> ## W1: Claims on "New Paradigm"
> Please refer to **Global Rebuttal G2** for a detailed discussion on video generation. We clarify two key points:
> 1.  **Paradigm Shift.** Our claim targets a pioneer shift in **methodology** rather than **task**—from **pixel-centric** (black-box denoising, e.g., Sora) to **code-centric** (white-box reasoning) generation, which enables editability and precise logic.
> 2.  **Task Definition.** Video Generation is defined by the **Text→Video** objective (refer to G2 [1-7]), regardless of using diffusion or code as the intermediate step.
>
> ## W2 & Q2: Validity of TeachQuiz
> Please refer to **Global Rebuttal G1** for our arguments on the necessity, rationale, and experimental validation of TeachQuiz.
>
> ## W3: Benchmark lacks domain diversity.
> We respectfully clarify several points that differ from our actual benchmark design.
> 1. **Domain Diversity.** MMMC is **not limited to math; it's multidisciplinary**. It contains **13 categories**, including physics, CS, diseases, economics, etc.
> 2. **Benchmark Availability.** Educational video generation is a new problem, and **no prior benchmark exists**. Ours represents the first systematic effort to define and evaluate this problem.
> 3. **Source Popularity.** 3Blue1Brown is not "niche". We chose it because it provides **high-quality videos with paired code**, and its **7.87M subscribers** reflect broad public interest.
> 4. **Generalization.** Robust performance on an additional Manim-animation benchmark (Appendix A.1.4) shows the approach is not tied to a single content source.
>
> ## W4: Pixel models are non-competitive.
> It is worth highlighting that our baseline selection was **carefully considered**.
> 1. **Comparison against strong models.** To ensure the comprehensive evaluation, our evaluation is **not limited** to pixel models. We also compare against and outperform baselines:
>    * **Strong Code LLMs:** Claude Opus 4.1 (a top-tier coding model).
>    * **Specialized Agents:** We outperform **TEA**, an existing Manim-based agent, by **+10 points** (Table 8).
> 2. **Pixel models are representative.** Models like Veo3 are the standard for general video generation. Including them is essential to demonstrate their limitations in handling educational structure, thereby motivating the shift to a code-centric approach.
>
> ## W5 & Q3: Feasibility, Generation Time, and Token Cost
> We clarify that the `43-minute` value is a misinterpretation.
> 1. **Actual Efficiency.** Our generation times are **8–16 minutes per 2-minute video**. Compared to the days required for manual professional production, our method has **an acceptable cost**.
>
>    |Setting|Time (w/o Parallel & SR)|Time|Tokens|Cost|
>    |-|-|-|-|-|
>    |Ours (GPT-5)|91.5 min|8.8 min|19.3K|$0.02|
>    |Ours (Claude Opus 4.1)|121.4 min|13.8 min|43.1K|$0.65|
>
> 2. **Scalability**. Our method supports different audiences, multiple domains, and flexible section lengths, enabling both short learning units and long-form videos.
>
> ## W6 & Q4: Reliability of the VLM-Critic
> We clarify that the need for a VLM-Critic is to **simulate human learning** and **actively identify the problem**.
>
> 1. **Simulating Human Learners.** We use the VLM-Critic as a scalable proxy for a human student, mimicking how learners perceive visual content. Humans also have subjective.
> 2. **Addressing Spatial Limitations**. We acknowledge that standard VLMs can struggle with precise layout details. To fix this, we propose Visual Anchor Prompting. This converts continuous visual cues into discrete grid references, enabling the Critic to reliably detect occlusions and layout errors (Tables 2 & 7) that raw VLMs might miss.
> 3. **Verified Human and VLM Alignment.** The validity is confirmed by data: strong correlations with human (**Aesthetics r=0.75; TeachQuiz r=0.69**) prove that our enhanced Critic accurately reflects real human preferences.
>
> ## Q1: Novelty of Code2Video
>
> 1. **New Problem Formulation.** We propose code-centric educational video generation, prioritizing executability and interpretability over pixel-level fidelity—a necessary shift for precise instructional content.
> 2. **Visual-centric Coding Agentic Designs.** To address VLMs' spatial imprecision, we convert continuous 2D reasoning into **discrete anchor selection** managed by an occupancy table. This ensures precise, conflict-free placement—a capability missing in existing agents.
> 3. **New Evaluation Perspective.** TeachQuiz introduces an **unlearn → watch → re-learn** metric to measure knowledge transfer, not just visual appearance. This aligns evaluation with the actual purpose of educational videos-a critical aspect unexplored in prior work.
> 4. **Significant Social Impact for AI Education.** Our framework offers a practical path for scalable, high-quality AI-generated instructional content.

---

> ### Author Response · Authors · 2025-11-25
>
> Dear Reviewer NpjQ,
>
> Thank you once again for your feedback!
>
> We would greatly appreciate it if you could review our response to ensure it adequately addresses your concerns. We remain fully dedicated to clarifying any remaining points and would welcome any further discussion to ensure all your questions are thoroughly answered.
>
> Thank you for your time and consideration.
>
> Best,
>
> Authors of 1990

---

> > ### Comment · Area_Chair_oDPp · 2025-11-26
> >
> > Dear reviewer NpjQ,
> >
> > Could you please respond to the author's feedback?
> >
> > AC

---

> ### Comment · Reviewer_NpjQ · 2025-11-26
> **RE: Rebuttal**
>
> Thank you for the response. I think my concerns have been addressed, and I will rise my score.

---

### Official Review · Reviewer_KLFQ · 2025-11-01

**Soundness:** 2
**Presentation:** 3
**Contribution:** 3
**Rating:** 6
**Confidence:** 3

**Summary:**

This paper proposes a code-centric agent framework for generating educational videos via executable Python code.

**Strengths:**

1. The paper is well organized.
2. This paper introduces a code-centric paradigm for educational video generation, positioning executable code as the unifying medium for temporal sequencing and spatial organization. This seems to be able to generate longer videos than Veo3.

**Weaknesses:**

1. The paper's task seems to be quite similar to PPT generation. Could you discuss the differences between them?
2. Where did you collect your data from?
3. The generated video is still not long enough; real educational videos are usually over 30 minutes.
4. Large language models suffer from hallucinations, and the knowledge they provide may not necessarily be accurate, which is a fatal weakness in teaching.
5. In Table 1, the author's method is six times slower than Veo3. Generating a 40-minute educational video is estimated to take 5 days, which makes the efficiency unacceptable.

**Questions:**

see weakness

**Details Of Ethics Concerns:**

No Ethics Concerns

---

> ### Author Response · Authors · 2025-11-20
> **Response to R1 KLFQ**
>
> Thank you for your thoughtful review and for recognizing the value of our code-centric paradigm for generating structured educational videos. We appreciate the opportunity to clarify your concerns.
>
> ## W1: Difference between PPT and Educational Video Generation
>
> For clearer comparison, we provide anonymous links to video samples and PPT comparisons: https://anonymous.4open.science/r/Code2Video-1990/README.md.
>
> Our task differs from PPT generation in three key aspects:
> 1. **Dynamic transition vs. Static presentation.** PPTs are static slide sequences, while our method generates fully scripted animations with continuous timing, smooth transitions, and object motion.
> 2. **Flexible Visual Design vs. Fixed Layouts.** PPTs operate under fixed layouts and limited interactivity, whereas our Manim-based pipeline supports arbitrary visual compositions, multi-entity motion, and fine-grained spatial control.
> 3. **Visual-centric vs. Text-centric Explanation.** PPTs often rely on text to convey information, with images as support. Our task centers on **visual** reasoning—using animated diagrams to explain concepts that text alone cannot express (e.g., geometric deformation, dynamic systems).
>
> ## W2: Data Collection Source
>
> Our dataset is sourced from the **3Blue1Brown (3B1B) YouTube corpus**, a widely recognized source of high-quality educational content known for its expert use of Manim. All videos are used with the **explicit awareness**. Notably, we have contacted the 3B1B team and **official obtained their permission regarding the video use within our benchmark**.
>
> ## W3: Generated videos are shorter than 30-minute lectures.
>
> Our choice of ~5-minute videos is motivated by **benchmarking practicality**, not a *limitation of the method*.
> 1. **Feasibility for Existing Models.** We note that generating a 5-minute video is already computationally intensive for our baseline. Thus, we chose 5 minutes to facilitate **easier comparison of generation performance within the community**.
> 2. **Availability for Future Work.** Despite MMMC provides both **short clips (~2 min)** for standardized evaluation and the full 40-minute reference videos from 3B1B. While generating long videos is promising, it remains challenging for existing methods. We thus leave it to future work.
> 3. **Scalability of Code2Video.** Our approach naturally supports long-form generation via **section-based composition**, **parallel code generation**, and **consistent layout control**. Multiple sections can be combined into 30–40 minute videos while preserving temporal and spatial coherence.
>
> ## W4: LLM Hallucination
> Thank you for highlighting this concern. Ensuring factual correctness is central to our approach.
> 1. **Minimal Hallucination on Easy Topics.** The learning topics covered (e.g., SVM, attention) are well-established and reliably handled by modern LLMs; the primary challenge is **effective teaching**, not factual recall.
> 2. **Error Type Analysis.** We manually verified 50 randomly sampled videos with the following accuracy. This demonstrates a very low factual error rate (4% overall), confirming **high reliability** for educational use. Thus, hallucination is not a concern.
>     |Metric|Accuracy|
>     |-|-|
>     |Formula|100%|
>     |Definition|94%|
>     |Example|96%|
>     |Overall|96%|
> 3. **Agent Designs for Safeguards.** Our approach mitigates hallucination by agent design:
>     * Planner: Uses the **External Database** containing verified reference images and **structured outlines** specifying definitions and examples.
>     * Coder: Generates **explicit, deterministic code**. Unlike pixel-based models, the output (formulas, diagrams) is transparent and inherently checkable, making errors easy to detect and fix.
>
> ## W5: Efficiency Comparison with Veo3
>
> **We respectfully clarify a misunderstanding** ("In Table 1, the author's method is six times slower than Veo3"). The raw generation times in Table 1 correspond to **videos of different durations**, making direct comparison misleading. When normalizing by video length, the actual efficiency is as follows. Our approach is therefore **2.5–3.5× faster** than Veo3 in terms of video-length-normalized efficiency.
> |Method|Duration (min)|Cost Time (min)|Efficiency|
> |-|-|-|-|
> |Veo3|0.13 (8 s)|2.3|17.3|
> |Ours (GPT-5)|1.8|8.8|4.9|
> |Ours (Claude Opus 4.1)|2.0|13.8|6.9|
>
> **Scalability Stress Test.** We further validated scalability by generating a full **40-minute educational video**. With parallel scheduling and our ScopeRefine local debugging strategy, the full pipeline runs **4.2× faster than without these optimizations**, producing the complete 40-minute video in **~1.5 hours**, demonstrating practical operational efficiency.

---

> > ### Comment · Reviewer_KLFQ · 2025-11-21
> > **Response**
> >
> > Thank you for your response. The author has addressed some of my concerns, and I maintain my initial positive score.

---

> > > ### Author Response · Authors · 2025-11-21
> > > **Thanks for your time and efforts!**
> > >
> > > We sincerely thank you for your continued engagement and for maintaining a positive assessment of our work. We are glad to hear that our responses have effectively addressed your concerns.
> > >
> > > We remain fully dedicated to clarifying any remaining points and would welcome any further discussion to ensure all your questions are thoroughly answered.
> > >
> > > If you feel that our rebuttal and revisions have strengthened the contribution of the paper, we would be truly grateful if you might consider adjusting your score to champion its acceptance.

---

### Author Response · Authors · 2025-11-20
**Global Author Rebuttal**

## General Summary
We sincerely thank all the reviewers for their time and constructive feedback. We are encouraged by the reviewers' recognition that:
* **Our work brings significant community contribution** (Lzn8, vi9d).
* **Our code-centric paradigm is clearly defined and practical** (KLFQ, NpjQ, vi9d).
* **Our evaluation metric is novel and well-justified** (NpjQ, vi9d).
* **Our paper is well-organized and easy to follow** (KLFQ, Lzn8).

Please find our detailed responses to your specific questions below. For clarity, we use the following notations:
* **G** - Global Response
* **W** – Weakness
* **Q** – Question

Before addressing individual questions, we want to highlight two aspects:

## G1: Unlearning in TeachQuiz
1. **Evaluation Focus.** Our goal is not to advance unlearning, but to use it as a tool to isolate and measure knowledge gained specifically from the video.
2. **Why In-context Unlearning on Closed-source Models?** It is necessary because: **(i) Selection of closed-source model.** Open-source VLMs show limitation to preceive education videos, requiring *strong closed-source models* for reliable evaluation. **(ii) In-context unlearning is practical.** With model parameters inaccessible, in-context unlearning becomes the only option, as per-question fine-tuning would be prohibitively expensive.
3. **How It Works & Why It's Effective.** In-context unlearning [1-3] suppresses specific knowledge without parameter updates. [3] shows that even *parameter training doesn't guarantee "forgetting" and is costly*. In-context methods, which often require prompt-based regularization, mainly change the default output sampling distribution, aiming to simulate the model's behaviors after forgetting. Our approach builds on this principle and demonstrates empirical effectiveness through:
   * Ablation studies (Table 6)
   * Strong Pearson correlation with 40 human scores on TeachQuiz ($r=0.69$)
4. **How does Training-based Unlearning Perform?** We compare against **fine-tuning [4] v.s. training-free [3]** unlearning methods. We adapt [4] by selecting 20 topics each for the forget and neighbor sets, fine-tuning the *llava-pretrain-llama-2-7b-chat* model. We generate videos using the forget set and test the TeachQuiz scores of methods [3], [4], and ours on these videos.
   * **Efficiency.** On LLaVA, the training-free approach **performs comparably** to fine-tuning while avoiding the high cost of training.
   * **Clearer TeachQuiz.** Training-free methods enable the use of **stronger closed-source models**. While fine-tuning is restricted to weaker open-source VLMs (yielding smaller learning gains), in-context unlearning leverages powerful models to demonstrate **clearer post-video improvement**.

|Model|Unlearning Method|Video by Claude Opus|Video by GPT-5|Video by GPT-4o|
|:-|:-|:-:|:-:|:-:|
|LLaVA|Fine-tuning [4]|30|24|15|
||In-Context [3]|32|25|16|
||**Ours**|31|20|13|
|GPT-5|In-Context [3]|70|61|33|
||**Ours**|72|61|36|
|Gemini-2.5 Pro| In-Context [3]|81|79|38|
||**Ours**|84|77|42|

[1] Can we edit factual knowledge by in-context learning, EMNLP 23

[2] In-Context Unlearning: Language Models as Few-Shot Unlearners, ICML 24

[3] Guardrail baselines for unlearning in llms, ICLRW 24

[4] A closer look at machine unlearning for large language models, ICLR 25

## G2: Is Code2Video under the scope of Video Generation?
1. **Definitions of Video Generation.** Modern research defines video generation as producing coherent videos from text [1,2], which exactly describes our work. The community recognizes **code-rendered and engine-generated methods as valid video generation** [3-5]. Early models producing simple motions were still considered video generators [6,7]—**the task is defined by input→output mapping, not by the underlying technical approach**.
2. **Diversity of Approaches.** Video generation has evolved through GANs, VAEs, AR, diffusion, and now hybrid or engine-assisted approaches. Our code-centric approach **offers a fresh perspective**: verifiability, editability, and precise layout control that are essential for educational content but challenging for pixel-based methods.
3. **Why Compare against Pixel Models?** Witness the recent progress by frontier video generation models. Pixel-based models are an intuitive starting point, but their limited control over long-term structure and spatial precision motivates our code-centric approach.

[1] CogVideo: Large-scale Pretraining for Text-to-Video Generation via Transformers, ICLR 23

[2] Imagen Video: High definition video generation with diffusion models, 2022

[3] Kubrick: Multimodal Agent Collaborations for Synthetic Video Generation, CVPRW 25

[4] Theoremexplainagent: Towards video-based multimodal explanations for LLM theorem understanding, ACL 25

[5] Exploring the Evolution of Physics Cognition in Video Generation: A Survey, 2025

[6] Video Generation From Text, AAAI 18

[7] To Create What You Tell: Generating Videos from Captions, MM 17

---

### Meta-Review · Area_Chair_Y3SV · 2026-01-05

**Summary:**

This paper presents Code2Video, an agentic pipeline (planner–coder–critic) that generates educational animations via executable Manim code, along with the MMMC benchmark and the TeachQuiz evaluation protocol. While the system is a reasonable engineering integration and the rebuttal clarifies some implementation/experimental details, the overall contribution does not meet the bar for ICLR.
The rebuttal addressed some issues (e.g., licensing clarification), but it does not resolve the core novelty and evaluation-validity objections. Given the mixed final scores and the remaining foundational concerns, I recommend reject.

**Reviewer Concerns:**

First, novelty and framing remain weak: as raised by the more negative reviewers (e.g., NpjQ and Lzn8), the work largely automates an existing Manim-based content creation workflow, and several claims around a “new paradigm on video generation” read overstated relative to the actual technical advance.
Second, the main empirical narrative relies heavily on TeachQuiz, but its “unlearning via prompting” premise is still controversial and may reflect instruction-following rather than genuine removal of prior knowledge, undermining confidence in the headline gains.
Third, concerns about generalization and verification remain: the benchmark style concentration and the lack of robust correctness checking for educational content limit the strength and reliability of the conclusions.

**Reviewer Scores:**

KLFQ (R1): No change (stays ~6). The rebuttal addresses several practical concerns (data and licensing clarification, pipeline details, some evaluation and efficiency discussion), but the core contribution level is unchanged, so I do not expect a meaningful score shift.
NpjQ (R2): Likely small increase, but still cautious. Even with additional clarifications, the main objections, overstated novelty, questionable robustness of TeachQuiz’s “unlearning,” and limited evidence of generalization, which are foundational and would likely prevent a full move to an above-threshold score.
Lzn8 (R3): No change (stays ~4). Their critique is primarily about problem definition and novelty (“video generation” vs. educational animation and content generation). The rebuttal may improve wording, but it does not materially change the underlying contribution in a way that would plausibly raise the score.
vi9d (R4): No change or very small change (stays ~2–3). Licensing concerns appear mitigated, but skepticism about the validity of the unlearning-based evaluation and the lack of strong correctness verification remains, so a substantial increase is unlikely.

---

### Decision · Program_Chairs · 2026-01-26

Reject